# Modeling resource allocation strategies for insecticide-treated bed nets to achieve malaria eradication

**Nora Schmit**[1]*[†], **Hillary M Topazian**[1†], **Matteo Pianella**[1†], **Giovanni D Charles**[1], **Peter Winskill**[1], **Michael T White**[2], **Katharina Hauck**[3], **Azra C Ghani**[1]

[1]MRC Centre for Global Infectious Disease Analysis, Imperial College London, London, United Kingdom; [2]Infectious Disease Epidemiology and Analytics G5 Unit, Department of Global Health, Institut Pasteur, Université de Paris, Paris, France; [3]MRC Centre for Global Infectious Disease Analysis, Jameel Institute, Imperial College London, London, United Kingdom

**\*For correspondence:**
n.schmit17@ic.ac.uk

[†]These authors contributed equally to this work

**Competing interest:** The authors declare that no competing interests exist.

**Abstract** Large reductions in the global malaria burden have been achieved, but plateauing funding poses a challenge for progressing towards the ultimate goal of malaria eradication. Using previously published mathematical models of *Plasmodium falciparum* and *Plasmodium vivax* transmission incorporating insecticide-treated nets (ITNs) as an illustrative intervention, we sought to identify the global funding allocation that maximized impact under defined objectives and across a range of global funding budgets. The optimal strategy for case reduction mirrored an allocation framework that prioritizes funding for high-transmission settings, resulting in total case reductions of 76% and 66% at intermediate budget levels, respectively. Allocation strategies that had the greatest impact on case reductions were associated with lesser near-term impacts on the global population at risk. The optimal funding distribution prioritized high ITN coverage in high-transmission settings endemic for *P. falciparum* only, while maintaining lower levels in low-transmission settings. However, at high budgets, 62% of funding was targeted to low-transmission settings co-endemic for *P. falciparum* and *P. vivax*. These results support current global strategies to prioritize funding to high-burden *P. falciparum*-endemic settings in sub-Saharan Africa to minimize clinical malaria burden and progress towards elimination, but highlight a trade-off with 'shrinking the map' through a focus on near-elimination settings and addressing the burden of *P. vivax*.

## eLife assessment

This study presents a **valuable** finding on the optimal prioritization in different malaria transmission settings for the distribution of insecticide-treated nets to reduce the malaria burden. The evidence supporting the claims of the authors is **solid**. The work will be of interest from a global funder perspective, though somewhat less relevant for individual countries.

## Introduction

Global support for malaria eradication has fluctuated in response to changing health policies over the past 75 years. From near global endemicity in the 1900's over 100 countries have eliminated malaria, with 10 of these certified malaria-free by the World Health Organization (WHO) in the last two decades (*Feachem et al., 2010*; *Shretta et al., 2017*; *Weiss et al., 2019*). Despite this success, 41% and 57% of the global population in 2017 were estimated to live in areas at risk of infection with *Plasmodium falciparum* and *Plasmodium vivax*, respectively (*Weiss et al., 2019*; *Battle et al., 2019*). In

2021 there were an estimated 247 million new malaria cases and over 600,000 deaths, primarily in children under 5 years of age (*World Health Organization, 2007*; *World Health Organization, 2022b*). Mosquito resistance to the insecticides used in vector control, parasite resistance to both first-line therapeutics and diagnostics, and local active conflicts continue to threaten elimination efforts (*World Health Organization, 2007*; *World Health Organization, 2022a*). Nevertheless, the global community continues to strive towards the ultimate aim of eradication, which could save millions of lives and thus offer high returns on investment (*Chen et al., 2018*; *Strategic Advisory Group on Malaria Eradication, 2020*).

The global goals outlined in the Global Technical Strategy for Malaria (GTS) 2016–2030 include reducing malaria incidence and mortality rates by 90%, achieving elimination in 35 countries, and preventing re-establishment of transmission in all countries currently classified as malaria-free by 2030 (*World Health Organization, 2007*; *World Health Organization, 2015*). Various stakeholders have also set timelines for the wider goal of global eradication, ranging from 2030–2050 (*World Health Organization, 2007*; *World Health Organization, 2020*, *Chen et al., 2018*; *Strategic Advisory Group on Malaria Eradication, 2020*). However, there remains a lack of consensus on how best to achieve this longer-term aspiration. Historically, large progress was made in eliminating malaria mainly in lower-transmission countries in temperate regions during the Global Malaria Eradication Program in the 1950s, with the global population at risk of malaria reducing from around 70% of the world population in 1950 to 50% in 2000 (*Hay et al., 2004*). Renewed commitment to malaria control in the early 2000s with the Roll Back Malaria initiative subsequently extended the focus to the highly endemic areas in sub-Saharan Africa (*Feachem et al., 2010*). Whilst it is now widely acknowledged that the current tool set is insufficient in itself to eradicate the parasite, there continues to be debate about how resources should be allocated (*Snow, 2015*). Some advocate for a focus on high-burden settings to lower the overall global burden (*World Health Organization, 2007*; *World Health Organization, 2019*), while others call for increased funding to middle-income low-burden countries through a 'shrink the map strategy' where elimination is considered a driver of global progress (*Newby et al., 2016*). A third set of policy options is influenced by equity considerations including allocating funds to achieve equal allocation per person at risk, equal access to bed nets and treatment, maximize lives saved, or to achieve equitable overall health status (*World Health Organization, 2007*; *World Health Organization, 2013*, *Raine et al., 2016*).

Global strategies are influenced by international donors, which represent 68% of the global investment in malaria control and elimination activities (*World Health Organization, 2007*; *World Health Organization, 2022b*). The Global Fund and the U.S. President's Malaria Initiative are two of the largest contributors to this investment. Their strategies pursue a combination approach, prioritizing malaria reduction in high-burden countries while achieving sub-regional elimination in select settings (*The Global Fund, 2021*, *United States Agency for International Development & Centers for Disease Control and Prevention, 2021*). Given that the global investment for malaria control and elimination still falls short of the 6.8 billion USD currently estimated to be needed to meet GTS 2016–2030 goals (*World Health Organization, 2007*; *World Health Organization, 2022b*), an optimized strategy to allocate limited resources is critical to maximizing the chance of successfully achieving the GTS goals and longer-term eradication aspirations.

In this study, we use mathematical modeling to explore the optimal allocation of limited global resources to maximize the long-term reduction in *P. falciparum* and *P. vivax* malaria. Our aim is to determine whether financial resources should initially focus on high-transmission countries, low-transmission countries, or a balance between the two across a range of global budgets. In doing so, we consider potential trade-offs between short-term gains and long-term impact. We use compartmental deterministic versions of two previously developed and tested individual-based models of *P. falciparum* and *P. vivax* transmission, respectively (*Griffin et al., 2010*; *White et al., 2018*). Using the compartmental model structures allows us to fully explore the space of possible resource allocation decisions using optimization, which would be prohibitively costly to perform using more complex individual-based models. Furthermore, to evaluate the impact of resource allocation options, we focus on a single intervention - insecticide-treated nets (ITNs). Whilst in reality, national malaria elimination programs encompass a broad range of preventative and therapeutic tools alongside different surveillance strategies as transmission decreases, this simplification is made for computational feasibility, with ITNs chosen as they (a) provide both an individual protective effect and population-level

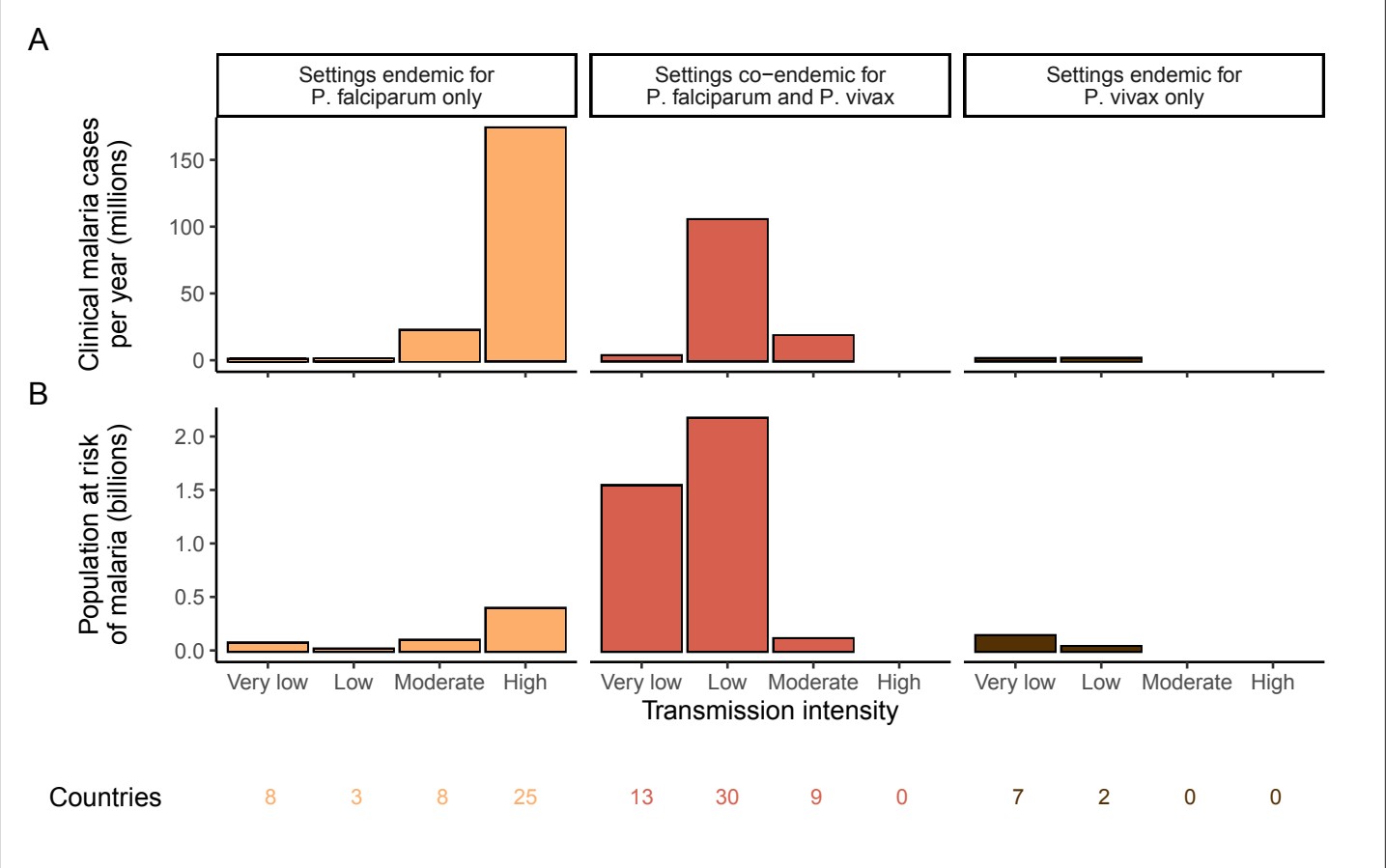

**Figure 1.** Global distribution of *P. falciparum* and *P. vivax* malaria burden in 2000 (in the absence of insecticide-treated nets) obtained from the Malaria Atlas Project (**Weiss et al., 2019**; **Battle et al., 2019**). (**A**) The annual number of clinical cases and (**B**) the population at risk of malaria across settings with different transmission intensities and endemic for *P. falciparum*, *P. vivax,* or co-endemic for both species. The number of countries in each setting is indicated below the figure.

transmission reductions (i.e. indirect effects); (b) are the most widely used single malaria intervention other than first-line treatment; and (c) extensive distribution and costing data are available that allow us to incorporate their decreasing technical efficiency at high coverage.

## Results

We identified 105 malaria-endemic countries based on 2000 *P. falciparum* and *P. vivax* prevalence estimates (before the scale-up of interventions), of which 44, 9, and 52 were endemic for *P. falciparum* only, *P. vivax* only, and co-endemic for both species, respectively. Globally, the clinical burden of malaria was focused in settings of high transmission intensity endemic for *P. falciparum* only, followed by low-transmission settings co-endemic for *P. falciparum* and *P. vivax* (**Figure 1A**). Conversely, 89% of the global population at risk of malaria was located in co-endemic settings with very low and low transmission intensities (**Figure 1B**). All 25 countries with high transmission intensity and 11 of 17 countries with moderate transmission intensity were in Africa, while almost half of global cases and populations at risk in low-transmission co-endemic settings originated in India.

Deterministic compartmental versions of two previously published and validated mathematical models of *P. falciparum* and *P. vivax* malaria transmission dynamics (**Griffin et al., 2010**; **Griffin et al., 2014**; **Griffin et al., 2016**; **White et al., 2018**) were used to explore associations between ITN use and clinical malaria incidence. In model simulations, the relationship between ITN usage and malaria infection outcomes varied by the baseline entomological inoculation rate (EIR), representing local transmission intensity, and parasite species (**Figure 2**). The same increase in ITN usage achieved a larger

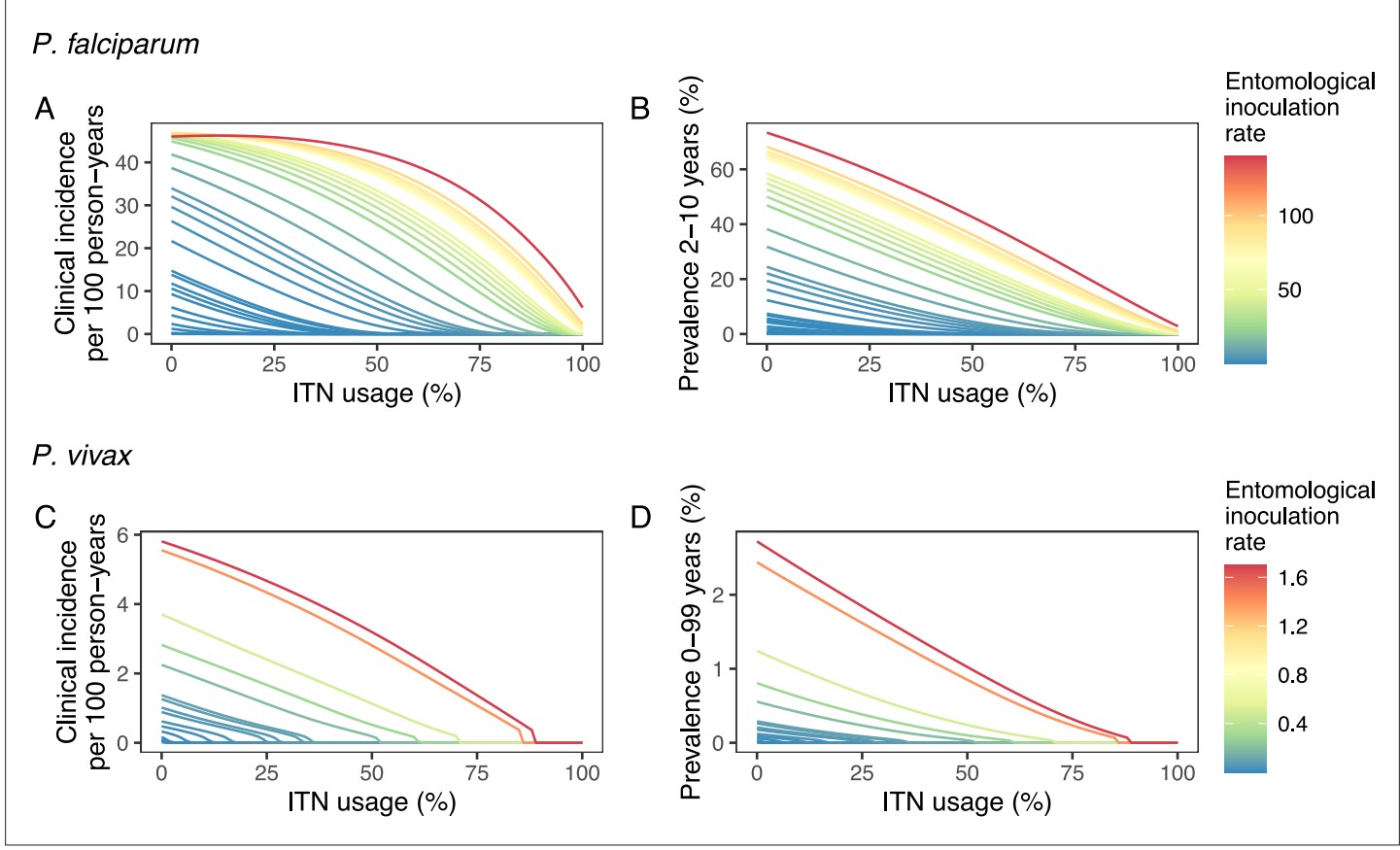

**Figure 2.** Modeled impact of insecticide-treated net (ITN) usage on malaria epidemiology by the setting-specific transmission intensity, represented by the baseline entomological inoculation rate. The impact on the clinical incidence and prevalence of *P. falciparum* malaria (panels **A** and **B**) and on the clinical incidence and prevalence of *P. vivax* malaria (panels **C** and **D**) is shown. Panels **A** and **C** represent the clinical incidence for all ages.

relative reduction in clinical incidence in low-EIR than in high-EIR settings. Low levels of ITN usage were sufficient to eliminate malaria in low-transmission settings, whereas high ITN usage was necessary to achieve a substantial decrease in clinical incidence in high-EIR settings. At the same EIR value, ITNs also led to a larger relative reduction in *P. falciparum* than *P. vivax* clinical incidence. However, ITN usage of 80% was not sufficient to lead to the full elimination of either *P. falciparum* or *P. vivax* in the highest transmission settings. In combination, the models projected that ITNs could reduce global *P. falciparum* and *P. vivax* cases by 83.6% from 252.0 million and by 99.9% from 69.3 million in 2000, respectively, assuming a maximum ITN usage of 80%.

We next used a non-linear generalized simulated annealing function to determine the optimal global resource allocation for ITNs across a range of budgets. We defined optimality as the funding allocation across countries which minimizes a given objective. We considered two objectives: first, reducing the global number of clinical malaria cases, and second, reducing both the global number of clinical cases and the number of settings not having yet reached a pre-elimination phase. The latter can be interpreted as accounting for an additional positive contribution of progressing towards elimination on top of a reduced case burden (e.g. general health system strengthening through a reduced focus on malaria). To relate funding to the impact on malaria, we incorporated a non-linear relationship between costs and ITN usage, resulting in an increase in the marginal cost of ITN distribution at high coverage levels (*Bertozzi-Villa et al., 2021*). We considered a range of fixed budgets, with the maximum budget being that which enabled achieving the lowest possible number of cases in the model. Low, intermediate, and high budget levels refer to 25%, 50%, and 75% of this maximum, respectively.

In our main analysis, we ignored the time dimension over which funds are distributed, instead focusing on the endemic equilibrium reached for each level of allocation (sensitivity to this assumption

is explored in a second analysis with dynamic re-allocation every 3 years). The optimal strategies were compared with three existing approaches to resource allocation: (1) prioritization of high-transmission settings, (2) prioritization of low-transmission (near-elimination) settings, and (3) proportional allocation by disease burden. Strategies prioritizing high- or low-transmission settings involved the sequential allocation of funding to groups of countries based on their transmission intensity (from highest to lowest EIR or *vice versa*). The proportional allocation strategy mimics the current allocation algorithm employed by the Global Fund: budget shares are distributed according to the malaria disease burden in the 2000–2004 period (*The Global Fund, 2019*). To allow comparison with this existing funding model, we also started allocation decisions from the year 2000.

We found that the optimal strategies for reducing total malaria cases (i.e. global burden) and for case reduction and pre-elimination to be similar to the strategy that prioritized funding for

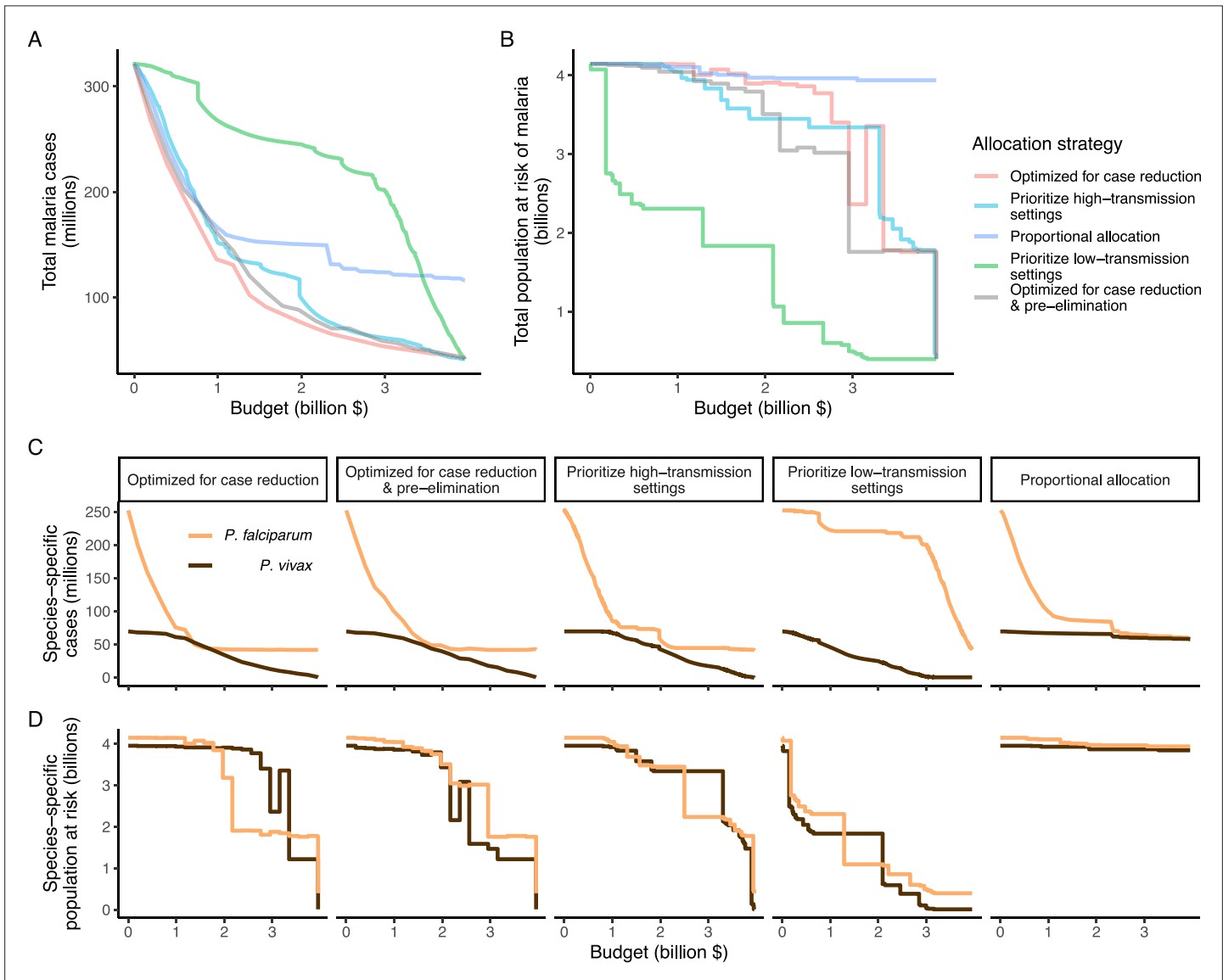

**Figure 3.** Global clinical cases and population at risk of malaria under different allocation strategies at varying budgets. The impact on total malaria cases (panel **A**), total population at risk (panel **B**), individual *P. falciparum* and *P. vivax* cases (panel **C**), and population at risk of either species (panel **D**) are shown. Budget levels range from 0, representing no usage of insecticide-treated nets, to the budget required to achieve the maximum possible impact. Optimizing for case reduction generally leads to declining populations at risk as the budget increases, but this is not guaranteed due to the possibility of redistribution of funding between settings to minimize cases. The strategy optimizing case reduction and pre-elimination shown here places the same weighting (1:1) on reaching pre-elimination in a setting as on averting total cases, but conclusions were the same for weights of 0.5–100 on pre-elimination.

high-transmission settings. These three strategies achieved the largest reductions in global malaria cases at all budgets, including reductions of 76%, 73%, and 66% at the intermediate budget level, respectively (*Figure 3A*, *Table 1*). At low to intermediate budgets, the proportional allocation strategy also reduced malaria cases effectively by up to 53%. While these four scenarios had very similar effects on malaria cases at low budgets, they diverged with increasing funding, where the proportional allocation strategy did not achieve substantial further reductions. Depending on the available budget, the optimal strategy for case reduction averted up to 31% more cases than prioritization of high-transmission settings and 64% more cases than proportional allocation, corresponding to respective differences of 37.9 and 74.5 million cases globally.

We additionally found there to be a trade-off between reducing global cases and reducing the global population at risk of malaria. Both the optimal strategies and the strategy prioritizing high-transmission settings did not achieve substantial reductions in the global population at risk until large investments were reached (*Figure 3B*, *Table 1*). Even at a high budget, the global population at risk was only reduced by 19% under the scenario prioritizing high-transmission settings, with higher reductions of 42–58% for the optimal strategies, while proportional allocation had almost no effect on this outcome. Conversely, diverting funding to prioritize low-transmission settings was highly effective at increasing the number of settings eliminating malaria, achieving a 56% reduction in the global

**Table 1.** Relative reduction in malaria cases and population at risk under different allocation strategies.

Reductions are shown relative to the baseline of 321 million clinical cases and 4.1 billion persons at risk in the absence of interventions. Low, intermediate, and high budget levels represent 25%, 50%, and 75% of the maximum budget, respectively. The strategy optimizing case reduction and pre-elimination shown here places the same weighting (1:1) on reaching pre-elimination in a setting as on averting total cases.

| | | Clinical cases | | Population at risk | |
|---|---|---|---|---|---|
| Scenario | Budget level | Number (millions) | Relative reduction (%) | Number (billions) | Relative reduction (%) |
| | Low | 136.1 | 58 | 4.1 | 0 |
| | Intermediate | 77.0 | 76 | 3.9 | 6 |
| | High | 53.9 | 83 | 2.4 | 42 |
| Optimized for case reduction | Maximum | 41.5 | 87 | 0.4 | 91 |
| | Low | 161.3 | 50 | 4.0 | 3 |
| | Intermediate | 87.8 | 73 | 3.5 | 16 |
| Optimized for case reduction & pre-elimination | High | 58.8 | 82 | 1.8 | 58 |
| | Maximum | 41.5 | 87 | 0.4 | 91 |
| | Low | 153.9 | 52 | 4.0 | 2 |
| | Intermediate | 109.5 | 66 | 3.4 | 17 |
| | High | 61.8 | 81 | 3.3 | 19 |
| Prioritize high-transmission settings | Maximum | 41.5 | 87 | 0.4 | 91 |
| | Low | 166.9 | 48 | 4.1 | 1 |
| | Intermediate | 150.4 | 53 | 4.0 | 4 |
| | High | 123.8 | 61 | 4.0 | 4 |
| Proportional allocation | Maximum | 116.0 | 64 | 3.9 | 5 |
| | Low | 268.2 | 17 | 2.3 | 44 |
| | Intermediate | 245.2 | 24 | 1.8 | 56 |
| | High | 202.1 | 37 | 0.5 | 88 |
| Prioritize low-transmission settings | Maximum | 41.5 | 87 | 0.4 | 91 |

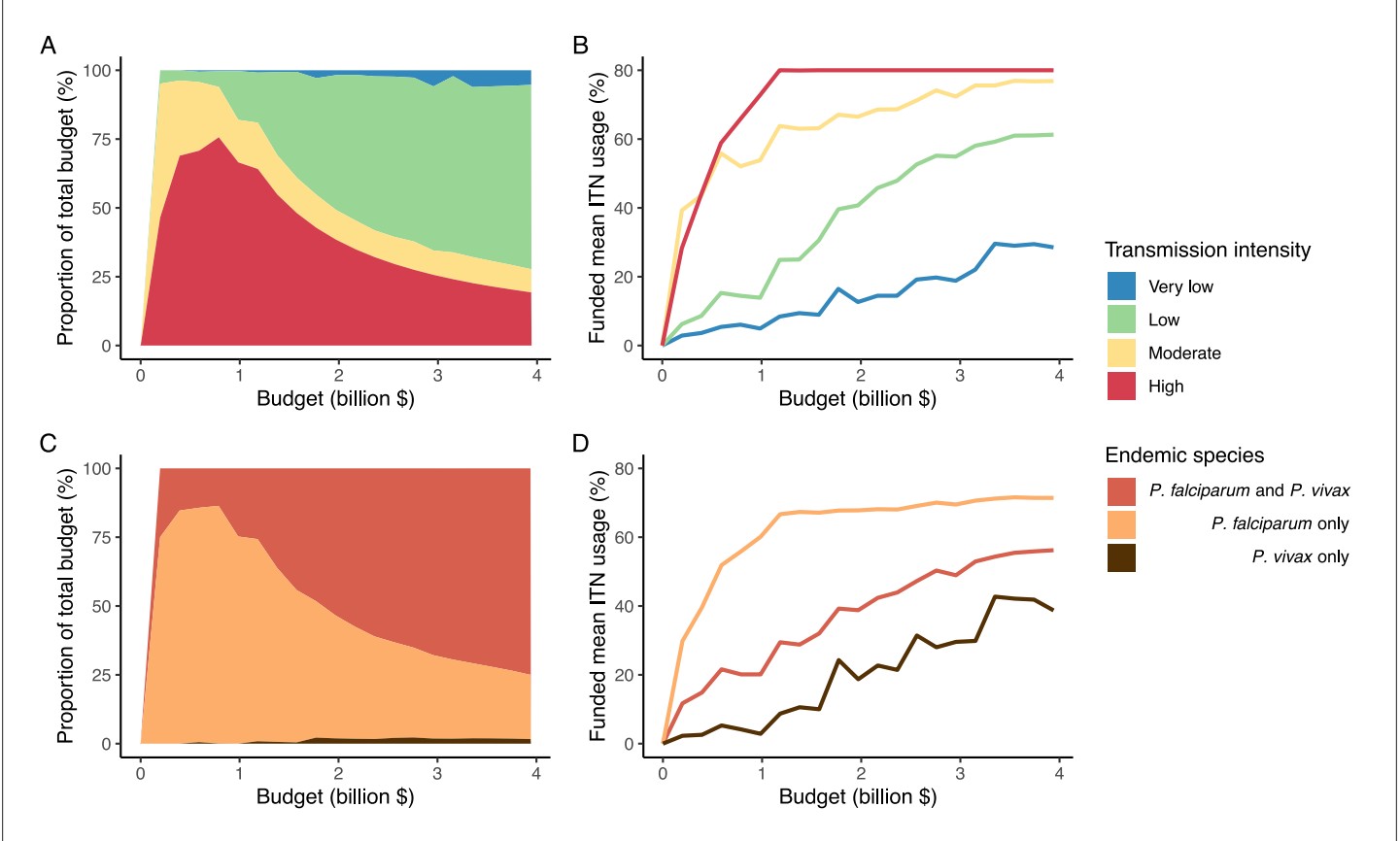

**Figure 4.** Optimal strategy for funding allocation across settings to minimize malaria case burden at varying budgets. Panels show optimized allocation patterns across settings of different transmission intensities (panels **A** and **B**) and different endemic parasite species (panels **C** and **D**). The proportion of the total budget allocated to each setting (panels **A** and **C**) and the resulting mean population usage of insecticide-treated nets (ITNs) (panels **B** and **D**) are shown.

population at risk already at intermediate budgets. However, this investment only led to a minimal reduction of 24% in total malaria case load (*Figure 3*, *Table 1*). At high budget levels, prioritizing low-transmission settings resulted in up to 3.8 times (a total of 159.4 million) more cases than the optimal allocation for case reduction. Despite the population at risk remaining relatively large with the optimal strategy for case reduction and pre-elimination, it nevertheless led to pre-elimination in more malaria-endemic settings than all other strategies (*Appendix 1—figure 7*), in addition to close to minimum cases across all budgets (*Figure 3*).

The allocation strategies also had differential impacts on *P. falciparum* and *P. vivax* cases, with case reductions generally occurring first for *P. falciparum* except when prioritizing low-transmission settings. *P. vivax* cases were not substantially affected at low global budgets for all other allocation strategies, and proportional allocation had almost no effect on reducing *P. vivax* clinical burden at any budget (*Figure 3C*), leading to a temporary increase in the proportion of total cases attributable to *P. vivax* relative to *P. falciparum*. The global population at risk remained high with the optimal strategy for case reduction even at high budgets, partly due to a large remaining population at risk of *P. vivax* infection (*Figure 3D*), which was not targeted when aiming to minimize total cases (*Figure 1*).

The optimized distribution of funding to minimize clinical burden depended on the available global budget and was driven by the setting-specific transmission intensity and the population at risk (*Figure 4*, *Figure 1*). With very low to low budget levels, as much as 85% of funding was allocated to moderate to high transmission settings (*Figure 4A*, *Appendix 1—figure 8A*). This allocation pattern led to the maximum ITN usage of 80% being reached in settings of high transmission intensity and smaller population sizes even at low budgets, while maintaining lower levels in low-transmission settings with larger populations (*Figure 4B*, *Appendix 1—figure 8B*). The proportion of the budget

allocated to low and very low transmission settings increased with increasing budgets, and low transmission settings received the majority of funding at intermediate to maximum budgets. This allocation pattern remained very similar when optimizing for both case reduction and pre-elimination (*Appendix 1—figure 9*). Similar patterns were also observed for the optimized distribution of funding between settings endemic for only *P. falciparum* compared to *P. falciparum* and *P. vivax* co-endemic settings (*Figure 4C–D*), with the former being prioritized at low to intermediate budgets. At the maximum budget, 70% of global funding was targeted at low- and very low-transmission settings co-endemic for both parasite species.

To evaluate the robustness of the results, we conducted a sensitivity analysis on our assumption of ITN distribution efficiency. Results remained similar when assuming a linear relationship between ITN usage and distribution costs (*Appendix 1—figure 10*). While the main analysis involves a single allocation decision to minimize long-term case burden (leading to a constant ITN usage over time in each setting irrespective of subsequent changes in burden), we additionally explored an optimal strategy with dynamic re-allocation of funding every 3 years to minimize cases in the short term. At high budgets, capturing dynamic changes over time through re-allocation of funding based on minimizing *P. falciparum* cases every 3 years led to the same case reductions over time as a one-time optimization with the allocation of a constant ITN usage (*Appendix 1—figure 11*). At lower budgets, re-allocation every 3 years achieved a higher impact at several timepoints, but total cases remained similar between the two approaches. Although reallocation of resources from settings which achieved elimination to higher transmission settings did not lead to substantially fewer cases, it reduced total spending over the 39 year period in some cases (*Appendix 1—figure 11*).

## Discussion

Our study highlights the potential impact that funding allocation decisions could have on the global burden of malaria. We estimated that optimizing ITN allocation to minimize global clinical incidence could, at a high budget, avert 83% of clinical cases compared to no intervention. In comparison, the optimal strategy to minimize the clinical incidence and maximize the number of settings reaching pre-elimination averted 82% of clinical cases, prioritizing high-transmission settings 81%, proportional allocation 61%, and prioritizing low-transmission settings 37%. Our results support initially prioritizing funding towards reaching high ITN usage in the high-burden *P. falciparum*- endemic settings to minimize global clinical cases and advance elimination in more malaria-endemic settings, but highlight the trade-off between this strategy and reducing the global population at risk of malaria as well as addressing the burden of *P. vivax*.

Prioritizing low-transmission settings demonstrated how focusing on 'shrinking the malaria map' by quickly reaching elimination in low-transmission countries diverts funding away from the high-burden countries with the largest caseloads. Prioritizing low-transmission settings achieved elimination in 42% of settings and reduced the global population at risk by 56% when 50% of the maximum budget had been spent, but also resulted in 3.2 times more clinical cases than the optimal allocation scenario. Investing a larger share of global funding towards high-transmission settings aligns more closely with the current WHO 'high burden to high impact' approach, which places an emphasis on reducing the malaria burden in the 11 countries which comprise 70% of global cases (*World Health Organization, 2007*; *World Health Organization, 2019*). Previous research supports this approach, finding that the 20 highest-burden countries would need to obtain 88% of global investments to reach case and mortality risk estimates in alignment with GTS goals (*Patouillard et al., 2017*). This is similar to the modeled optimized funding strategy presented here, which allocated up to 76% of very low budgets to settings of high transmission intensity located in sub-Saharan Africa. An initial focus on high- and moderate-transmission settings is further supported by our results showing that a balance can be found between achieving close to optimal case reductions while also progressing towards elimination in the maximum number of settings. Even within a single country, targeting interventions to local hot-spots has been shown to lead to higher cost savings than universal application (*Barrenho et al., 2017*), and could lead to elimination in settings where untargeted interventions would have little impact (*Bousema et al., 2012*).

Assessing optimal funding patterns is a global priority due to the funding gap between supply and demand for resources for malaria control and elimination (*World Health Organization, 2007*; *World Health Organization, 2022b*). However, allocation decisions will remain important even if more

funding becomes available, as some of the largest differences in total cases between the modeled strategies occurred at intermediate to high budgets. Our results suggest that most of global funding should only be focused in low-transmission settings co-endemic for *P. falciparum* and *P. vivax* at high budgets once ITN use has already been maximized in high-transmission settings. Global allocation decisions are likely to affect *P. falciparum* and *P. vivax* burden differently, which could have implications for the future global epidemiology of malaria. For example, with a focus on disease burden reduction, a temporary increase in the proportion of malaria cases attributable to *P. vivax* was projected, in line with recent observations in near-elimination areas (*Battle et al., 2019*; *Price et al., 2020*). Nevertheless, even when international funding for malaria increased between 2007–2009, African countries remained the major recipients of financial support, while *P. vivax*-dominant countries were not as well funded (*Snow et al., 2010*). This serves as a reminder that achieving the elimination of malaria from all endemic countries will ultimately require targeting investments so as to also address the burden of *P. vivax* malaria.

Different priorities in resource allocation decisions greatly affect which countries receive funding and what health benefits are achieved. The modeled strategies follow key ethical principles in the allocation of scarce healthcare resources, such as targeting those of greatest need (*prioritizing high-transmission settings*, *proportional allocation*) or those with the largest expected health gain (*optimized for case reduction*, *prioritizing high-transmission settings*) (*World Health Organization, 2007*; *World Health Organization, 2013*). Allocation proportional to disease burden did not achieve as great an impact as other strategies because the funding share assigned to settings was constant irrespective of the invested budget and its impact. In modeling this strategy, we did not reassign excess funding in high-transmission settings to other malaria interventions, as would likely occur in practice. This illustrates the possibility that such an allocation approach can potentially target certain countries disproportionally and result in further inequities in health outcomes (*Barrenho et al., 2017*). From an international funder perspective, achieving vertical equity might, therefore, also encompass higher disbursements to countries with lower affordability of malaria interventions (*Barrenho et al., 2017*), as reflected in the Global Fund's proportional allocation formula which accounts for the economic capacity of countries and specific strategic priorities (*The Global Fund, 2019*). While these factors were not included in the proportional allocation used here, the estimated impact of these two strategies was nevertheless very similar (*Appendix 1—figure 12*).

While our models are based on country patterns of transmission settings and corresponding populations in 2000, there are several factors leading to heterogeneity in transmission dynamics at the national and sub-national levels which were not modeled and limit our conclusions. Seasonality, changing population size, and geographic variation in *P. vivax* relapse patterns or in mosquito vectors could affect the projected impact of ITNs and optimized distribution of resources across settings. The two representative *Anopheles* species used in the simulations are also both very anthropophagic, which may have led to an overestimation of the effect of ITNs in some settings. By using ITNs as the sole means to reduce mosquito-to-human transmission, we did not capture the complexities of other key interventions that play a role in burden reduction and elimination, the geospatial heterogeneity in cost-effectiveness and optimized distribution of intervention packages on a sub-national level, or related pricing dynamics (*Conteh et al., 2021*; *Drake et al., 2017*). For *P. vivax* in particular, reducing the global economic burden and achieving elimination will depend on the incorporation of hypnozoitocidal treatment and G6PD screening into case management (*Devine et al., 2021*). Furthermore, for both parasites, intervention strategies generally become more focal as transmission decreases, with targeted surveillance and response strategies prioritized over widespread vector control. Therefore, policy decisions should additionally be based on analysis of country-specific contexts, and our findings are not informative for individual country allocation decisions. Results do, however, account for nonlinearities in the relationship between ITN distribution and usage to represent changes in cost as a country moves from control to elimination: interventions that are effective in malaria control settings, such as widespread vector control, may be phased out or limited in favor of more expensive active surveillance and a focus on confirmed diagnoses and at-risk populations (*Shretta et al., 2017*). We also assumed that transmission settings are independent of each other, and did not allow for the possibility of re-introduction of disease, such as has occurred throughout the Eastern Mediterranean from imported cases (*World Health Organization, 2007*). While our analysis presents allocation strategies to progress toward eradication, the results do not provide insight into the allocation of funding

to maintain elimination. In practice, the threat of malaria resurgence has important implications for when to scale back interventions.

Our analysis demonstrates the most impactful allocation of a global funding portfolio for ITNs to reduce global malaria cases. Unifying all funding sources in a global strategic allocation framework as presented here requires international donor allocation decisions to account for available domestic resources. National governments of endemic countries contribute 31% of all malaria-directed funding globally (*World Health Organization, 2020*), and government financing is a major source of malaria spending in near-elimination countries in particular (*Haakenstad et al., 2019*). Within the wider political economy which shapes the funding landscape and priority setting, there remains substantial scope for optimizing allocation decisions, including improving the efficiency of within-country allocation of malaria interventions. Subnational malaria elimination in localized settings within a country can also provide motivation for continued elimination in other areas and friendly competition between regions to boost global elimination efforts (*Lindblade and Kachur, 2020*). Although more efficient allocation cannot fully compensate for projected shortfalls in malaria funding, mathematical modeling can aid efforts in determining optimal approaches to achieve the largest possible impact with available resources.

## Materials and methods
### Transmission models
We used deterministic compartmental versions of two previously published individual-based transmission models of *P. falciparum* and *P. vivax* malaria to estimate the impact of varying ITN usage on clinical incidence in different transmission settings. The *P. falciparum* model has previously been fitted to age-stratified data from a variety of sub-Saharan African settings to recreate observed patterns in parasite prevalence ($PfPR_{2-10}$), the incidence of clinical disease, immunity profiles, and vector components relating to rainfall, mosquito density, and the EIR (*Griffin et al., 2016*). We developed a deterministic version of an existing individual-based model of *P. vivax* transmission, originally calibrated to data from Papua New Guinea but also shown to reproduce global patterns of *P. vivax* prevalence and clinical incidence (*White et al., 2018*). Models for both parasite species are structured by age and heterogeneity in exposure to mosquito bites, and account for human immunity patterns. They model mosquito transmission and population dynamics, and the impact of scale-up of ITNs in identical ways. Full assumptions, mathematical details, and parameter values can be found in Appendix 1 and in previous publications (*Griffin et al., 2010*; *Griffin et al., 2014*; *Griffin et al., 2016*; *White et al., 2018*).

### Data sources
We calibrated the model to baseline transmission intensity in all malaria-endemic countries before the scale-up of interventions, using the year 2000 as an indicator of these levels in line with the current allocation approach taken by the Global Fund (*The Global Fund, 2019*). Annual EIR was used as a measure of parasite transmission intensity, representing the rate at which people are bitten by infectious mosquitoes. We simulated models to represent a wide range of EIRs for *P. falciparum* and *P. vivax*. These transmission settings were matched to 2000 country-level prevalence data resulting in EIRs of 0.001–80 for *P. falciparum* and 0.001–1.3 for *P. vivax*. *P. falciparum* estimates came from parasite prevalence in children aged 2–10 years and *P. vivax* prevalence estimates came from light microscopy data across all ages, based on standard reporting for each species (*Weiss et al., 2019*; *Battle et al., 2019*). The relationship between parasite prevalence and EIR for specific countries is shown in *Appendix 1—figures 5 and 6*. In each country, the population at risk for *P. falciparum* and *P. vivax* malaria was obtained by summing WorldPop gridded 2000 global population estimates (*Tatem, 2017*) within Malaria Atlas Project transmission spatial limits using geoboundaries (*Runfola et al., 2020*) (Appendix 1: Country-level data and modeling assumptions on the global malaria distribution). The analysis was conducted on the national level, since this scale also applies to funding decisions made by international donors (*The Global Fund, 2019*). As this exercise represents a simplification of reality, population sizes were held constant, and projected population growth is not reflected in the number of cases and the population at risk in different settings. Seasonality was also not incorporated in the model, as EIRs are matched to annual prevalence estimates and the effects of seasonal changes

are averaged across the time frame captured. For all analyses, countries were grouped according to their EIR, resulting in a range of transmission settings compatible with the global distribution of malaria. Results were further summarized by grouping EIRs into broader transmission intensity settings according to WHO prevalence cut-offs of 0–1%, 1–10%, 10–35%, and ≥35% (*World Health Organization, 2007*; *World Health Organization, 2022a*). This corresponded approximately to classifying EIRs of less than 0.1, 0.1–1, 1–7, and 7 or higher as very low, low, moderate and high transmission intensity, respectively.

## Interventions

In all transmission settings, we simulated the impact of varying coverages of ITNs on clinical incidence. While most countries implement a package of combined interventions, to reduce the computational complexity of the optimization we considered the impact of ITN usage alone in addition to 40% treatment of clinical disease. ITNs are a core intervention recommended for large-scale deployment in areas with ongoing malaria transmission by WHO (*Winskill et al., 2019*; *World Health Organization, 2007*; *World Health Organization, 2022a*) and funding for vector control represents much of the global investments required for malaria control and elimination (*Patouillard et al., 2017*). Modeled coverages represent population ITN usage between 0 and 80%, with the upper limit reflective of common targets for universal access (*Koenker et al., 2018*). In each setting, the models were run until clinical incidence stabilized at a new equilibrium with the given ITN usage.

Previous studies have shown that, as population coverage of ITNs increases, the marginal cost of distribution increases as well (*Bertozzi-Villa et al., 2021*). We incorporated this non-linearity in costs by estimating the annual ITN distribution required to achieve the simulated population usage based on published data from across Africa, assuming that nets would be distributed on a 3-yearly cycle and accounting for ITN retention over time (Appendix 1). The cost associated with a given simulated ITN usage was calculated by multiplying the number of nets distributed per capita per year by the population size and by the unit cost of distributing an ITN, assumed to be $3.50 (*Sherrard-Smith et al., 2022*).

## Optimization

The optimal funding allocation for case reduction was determined by finding the allocation of ITNs $b$ across transmission settings that minimizes the total number of malaria cases at equilibrium. Case totals were calculated as the sum of the product of clinical incidence $cinc_i$ and the population $p_i$ in each transmission setting $i$. Simultaneous optimization for case reduction and pre-elimination was implemented with an extra weighting term in the objective function, corresponding to a reduction in total remaining cases by a proportion $w$ of the total cases averted by the ITN allocation, $C$. This, therefore, represents a positive contribution for each setting reaching the pre-elimination phase. The weighting on pre-elimination compared to case reduction was 0 in the scenario optimized for case reduction, and varied between 0.5 and 100 times in the other optimization scenarios. Resource allocation must respect a budget constraint, which requires that the sum of the cost of the ITNs distributed cannot exceed the initial budget $B$, with $b_i$ the initial number of ITNs distributed in setting $i$ and $c$ the cost of a single pyrethroid-treated net. The second constraint requires that the ITN usage $b_i^\star$ must be between 0 and 80% (*Koenker et al., 2018*), with ITN usage being a function of ITNs distributed, as shown in the following equation.

$$\min_{b \in R^n} \left[ \sum_i^n cinc_i * p_i \;-\; w * C * \sum_{i=1}^n j_i \right]$$

$$\text{s.t.} \qquad \sum_{i=1}^n b_i * c \leq B$$

$$0 \leq b_i^\star \leq 0.8 \ \ \forall \, i = 1, \ldots, n$$

$$C = Cases\ at\ baseline - \sum_i^n cinc_i * p_i$$

$$j_i = \begin{cases} 1, & cinc_i < 1/1000 \\ 0, & cinc_i \geq 1/1000 \end{cases}$$

$$b_i^\star = f(b_i)$$

$$\text{for all } i = 1, \ldots, n$$

The optimization was undertaken using generalized simulated annealing (*Xiang et al., 2013*). We included a penalty term in the objective function to incorporate linear constraints. Further details can be found in Appendix 1.

The optimal allocation strategy for minimizing cases was also examined over a period of 39 years using the *P. falciparum* model, comparing a single allocation of a constant ITN usage to minimize clinical incidence at 39 years, to reallocation every 3 years (similar to Global Fund allocation periods *The Global Fund, 2016*) leading to varying ITN usage over time. At the beginning of each 3 year period, we determined the optimized allocation of resources to be held fixed until the next round of funding, with the objective of minimizing 3 year global clinical incidence. Once *P. falciparum* elimination is reached in a given setting, ITN distribution is discontinued, and in the next period, the same total budget *B* will be distributed among the remaining settings. We calculated the total budget required to minimize case numbers at 39 years and compared the impact of re-allocating every 3 years with a one-time allocation of 25%, 50%, 75%, and 100% of the budget. To ensure computational feasibility, 39 years was used as it was the shortest time frame over which the effect of re-distribution of funding from countries having achieved elimination could be observed.

## Analysis

We compared the impact of the two optimal allocation strategies (scenarios 1 A and 1B) and three additional allocation scenarios on global malaria cases and the global population at risk. Modeled

**Table 2.** Overview of modeled scenarios for allocation of funding to different transmission settings.

Strategies 1A-1E compare resource allocation scenarios using clinical incidence values from each transmission setting at equilibrium after insecticide-treated net (ITN) coverage has been introduced. Strategies 2A-2B are compared as part of the allocation over time sub-analysis. EIR: entomological inoculation rate.

|  | Strategy | Modeling approach/assumptions |
|---|---|---|
| 1A | Optimized for total malaria case reduction | Generalized simulated annealing is used to determine the optimal allocation of a given budget to minimize the total number of global malaria cases. |
| 1B | Optimized for total malaria case reduction and pre-elimination | Generalized simulated annealing is used to determine the optimal allocation of a given budget to minimize the total number of global malaria cases while placing a premium on the pre-elimination phase being reached in a setting. |
| 1C | Prioritize high-transmission settings | Funding is allocated to groups of countries according to transmission intensity (*P. falciparum + P. vivax* entomological inoculation rate, EIR). For a given budget, the transmission settings with the highest EIR are prioritized, increasing ITN coverage in increments of 1% in each setting until malaria is eliminated or until an increase in coverage leads to no further decrease in cases, before allocating to the next-highest EIR setting. |
| 1D | Prioritize low-transmission (near-elimination) settings | Funding is allocated to groups of countries according to transmission intensity (*P. falciparum + P. vivax* EIR). For a given budget, the transmission settings with the lowest EIR are prioritized, increasing ITN coverage in increments of 1% in each setting until malaria is eliminated or until an increase in coverage leads to no further decrease in cases, before allocating to the next-lowest EIR setting. |
| 1E | Proportional allocation | Funding is allocated to groups of countries in proportion to their disease burden. Budget shares are calculated using country data from the World Malaria Report (*World Health Organization, 2007*; *World Health Organization, 2020*) and account for the country-specific total malaria cases (*P. falciparum* and *P. vivax*), deaths, incidence and mortality rate in 2000–2004, scaled by the subsequent increase in the population at risk (*The Global Fund, 2019*). |
| 2A | One-time optimized allocation for *P. falciparum* case reduction | Generalized simulated annealing is used to determine the optimized allocation at a given budget, minimizing the total number of global *P. falciparum* cases after 39 years, resulting in constant ITN usage in each setting over this time period. |
| 2B | Optimized allocation every three years for *P. falciparum* case reduction | Generalized simulated annealing is used to determine the optimized allocation at a given budget, minimizing the total number of global *P. falciparum* cases after every 3 year period for 39 years, allowing ITN usage to vary in each setting every 3 years. |

scenarios are shown in *Table 2*. Scenarios 1C-1E represent existing policy strategies that involve prioritizing high-transmission settings, prioritizing low-transmission (near-elimination) settings, or resource allocation proportional to disease burden in the year 2000. Global malaria case burden and the population at risk were compared between baseline levels in 2000 and after reaching an endemic equilibrium under each scenario for a given budget.

Certification of malaria elimination requires proof that the chain of indigenous malaria transmission has been interrupted for at least 3 years and a demonstrated capacity to prevent return transmission (*World Health Organization, 2007*; *World Health Organization, 2018*). In our analysis, transmission settings were defined as having reached malaria elimination once less than one case remained per the setting's total population. Once a setting reaches elimination, the entire population is removed from the global total population at risk, representing a 'shrink the map' strategy. The pre-elimination phase was defined as having reached less than 1 case per 1000 persons at risk in a setting (*Mendis et al., 2009*).

All strategies were evaluated at different budgets ranging from 0 to the minimum investment required to achieve the lowest possible number of cases in the model (noting that ITNs alone are not predicted to eradicate malaria in our model). No distinctions were made between national government spending and international donor funding, as the purpose of the analysis was to look at resource allocation and not to recommend specific internal and external funding choices.

All analyses were conducted in R v. 4.0.5 (R Foundation for Statistical Computing, Vienna, Austria). The sf (v. 0.9–8, *Pebesma, 2018*), raster (v. 3.4–10, *Hijmans and Van Etten, 2012*), and terra (v.1.3–4, *Hijmans et al., 2022*) packages were used for spatial data manipulation. The Akima package (v.0.6–2.2, *Akima et al., 2022*) was used for surface development, and the GenSA package (v.1.1.7, *Gubian et al., 2023*) for model optimization.

## Acknowledgements

This work was supported by the Wellcome Trust [reference 220900/Z/20/Z]. NS, HMT, MP, GDC, PW, KH, and ACG also acknowledge funding from the MRC Centre for Global Infectious Disease Analysis [reference MR/R015600/1], jointly funded by the UK Medical Research Council (MRC) and the UK Foreign, Commonwealth & Development Office (FCDO), under the MRC/FCDO Concordat agreement and is also part of the EDCTP2 program supported by the European Union. KH also acknowledges funding by Community Jameel. Disclaimer: 'The views expressed are those of the author(s) and not necessarily those of the NIHR, the UK Health Security Agency or the Department of Health and Social Care'. For the purpose of open access, the authors have applied a 'Creative Commons Attribution' (CC BY) license to any Author Accepted Manuscript version arising from this submission.

## Additional information

### Funding

| Funder | Grant reference number | Author |
| --- | --- | --- |
| Wellcome Trust | 10.35802/220900 | Nora Schmit<br>Hillary M Topazian<br>Matteo Pianella<br>Katharina Hauck<br>Azra C Ghani<br>Giovanni D Charles |
| Medical Research Council | MR/R015600/1 | Nora Schmit<br>Hillary M Topazian<br>Matteo Pianella<br>Giovanni D Charles<br>Peter Winskill<br>Katharina Hauck<br>Azra C Ghani |
| Community Jameel | | Katharina Hauck |

| Funder | Grant reference number | Author |
|--------|------------------------|--------|

The funders had no role in study design, data collection and interpretation, or the decision to submit the work for publication. For the purpose of Open Access, the authors have applied a CC BY public copyright license to any Author Accepted Manuscript version arising from this submission.

## Author contributions

Nora Schmit, Hillary M Topazian, Conceptualization, Data curation, Formal analysis, Investigation, Visualization, Methodology, Writing – original draft, Writing – review and editing; Matteo Pianella, Conceptualization, Data curation, Formal analysis, Investigation, Visualization, Methodology, Writing – review and editing; Giovanni D Charles, Conceptualization, Software, Methodology, Writing – review and editing; Peter Winskill, Software, Validation, Writing – review and editing; Michael T White, Validation, Writing – review and editing, Domain knowledge; Katharina Hauck, Azra C Ghani, Conceptualization, Supervision, Funding acquisition, Validation, Project administration, Writing – review and editing

## Author ORCIDs

Nora Schmit ⓘ https://orcid.org/0000-0001-9840-0878
Hillary M Topazian ⓘ http://orcid.org/0000-0001-7824-9605
Michael T White ⓘ http://orcid.org/0000-0002-7472-4138

Reviewer #1 (Public Review): https://doi.org/10.7554/eLife.88283.3.sa1
Reviewer #2 (Public Review): https://doi.org/10.7554/eLife.88283.3.sa2
Author Response https://doi.org/10.7554/eLife.88283.3.sa3

# Additional files

## Supplementary files

• MDAR checklist

## Data availability

The manuscript is a computational study, so no data have been generated. The previously published malaria transmission models code is available to download at GitHub (copy archived at *Unwin, 2023*). The code to conduct the analysis and produce the figures and tables in the manuscript are available to download at GitHub (copy archived at *Mrc-ide, 2022*). Datasets of parasite prevalence and spatial limits used in the analysis are publicly available from the Malaria Atlas Project at https://malariaatlas.org/.

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

# Appendix 1

## Mathematical models

### Overview

In this paper, we use an existing deterministic, compartmental, mathematical model of *P. falciparum* malaria transmission between humans and mosquitoes, which was originally calibrated to age-stratified data from settings across sub-Saharan Africa (*Griffin et al., 2016*). We also developed a deterministic version of an existing individual-based model of *P. vivax* transmission, originally calibrated to data from Papua New Guinea but also shown to reproduce global epidemiological patterns (*White et al., 2018*). Both models are structured by age and heterogeneity in exposure to mosquito bites, and allow for the presence of maternal immunity at birth and naturally acquired immunity across the life course. The mosquito and vector control components are modeled identically in both models, except for the force of infection acting on mosquitoes. A diagram of the model structures with human and adult mosquito components is shown in *Appendix 1-figure 1*.

Population and transmission dynamics were modeled separately for both species, assuming they are independent of each other, because the epidemiological significance of biological interactions between the parasites within hosts remains unclear (*Mueller et al., 2013*).

Note that while the term 'individuals' may be used in descriptions, the models are compartmental and do not track individuals; compartments represent the average number of people in a given state.

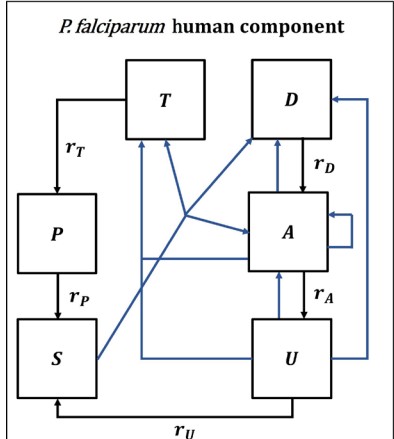
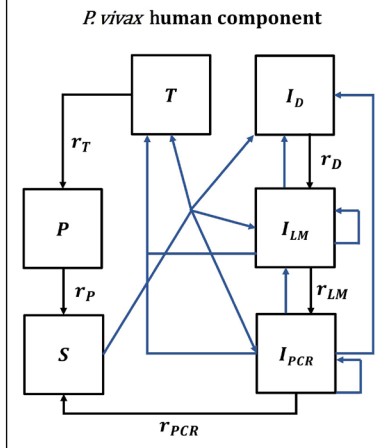
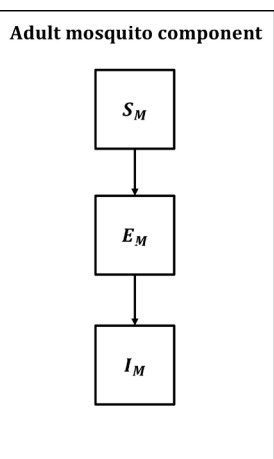

**Appendix 1—figure 1.** Malaria transmission model; diagram adapted from *Griffin et al., 2016* and *White et al., 2018*. Humans move through six states in the models for both species: *S* (susceptible), *D* (untreated symptomatic infection), *T* (successfully treated symptomatic infection), *A* (asymptomatic infection), *U* (asymptomatic sub-patent infection), and *P* (prophylaxis) in the *P. falciparum* model, and *S* (susceptible), $I_D$ (untreated symptomatic infection), *T* (successfully treated symptomatic infection), $I_{LM}$ (asymptomatic light microscopy detectable blood-stage infection), $I_{PCR}$ (asymptomatic sub-microscopic PCR detectable blood-stage infection) and *P* (prophylaxis) in the *P. vivax* model. New infections (including superinfections) are highlighted in blue but parameters are not shown. Rates $r_D$, $r_A$, $r_U$, $r_T$, $r_P$, $r_{LM}$, and $r_{PCR}$ determine the mean duration of each state. Hypnozoites states in the *P. vivax* model are not shown on the diagram. Adult female mosquitoes move through three model compartments: $S_m$ (susceptible), $E_m$ (exposed), and $I_m$ (infected).

### Human demography

In both the *P. falciparum* and *P. vivax* models, the aging process in the human population follows an exponential distribution. Humans can reach a maximum age of 100 years and experience a constant death rate of 1/21 per year based on the assumed median age of the population. The birth rate was assumed to equal the mortality rate so that the population remains stable over time. Demographic changes over time are, therefore, not accounted for.

In all following sections, human demographic processes are omitted from the equations of the transmission models and the immunity models for simplicity. All compartments experience the same constant mortality rate, while all births occur in the susceptible compartment.

## Heterogeneity in mosquito biting rates

In both models, exposure to mosquito bites is assumed to depend on age, due to varying body surface area and behavioral patterns.

The relative biting rate at age $a$ is calculated as:

$$\psi\left(a\right) = \left(1 - \rho \exp\left(-a/a_0\right)\right) \tag{1}$$

Where $\rho$ and $a_0$ are estimated parameters determining the relationship between age and biting rate.

Additionally, the human population in the model is stratified according to lifetime relative biting rate $\zeta$ which represents the heterogeneity in exposure to mosquito bites that occurs at various spatial scales, for example, due to attractiveness of humans to mosquitoes, housing standards and proximity to mosquito breeding sites, and is described by a log-normal distribution with a mean of 1, as follows:

$$\log\left(\zeta\right) \sim N\left(-\sigma^2/2, \sigma^2\right) \tag{2}$$

## *P. falciparum* human model component

In the *P. falciparum* model, humans move through four states of transmission and are present in only one of the six states at each timestep: susceptible ($S$), untreated symptomatic infection ($D$), successfully treated symptomatic infection ($T$), asymptomatic infection ($A$), asymptomatic sub-patent infection ($U$), and prophylaxis ($P$). Individuals in the model are born susceptible to infection but are temporarily protected by maternal immunity during the first six months of life. Humans are exposed to infectious bites from mosquitoes and are infected at a rate $\Lambda$, representing the force of infection from mosquitoes to humans. The force of infection depends on an individual's pre-erythrocytic immunity, the age-dependent biting rate, and the mosquito population size and level of infectivity. Following a latent period, $d_E$, and depending on clinical immunity levels, a proportion $\phi$ of infected individuals develop clinical disease, while the remaining move into the asymptomatic infection state. A proportion $f_T$ of those with clinical disease are successfully treated. Treated individuals recover from infection at the rate $r_T$ and move to the prophylaxis state, which represents a period of drug-dependent partial protection from reinfection. Recovery from untreated symptomatic infection to the asymptomatic infection state occurs at a rate $r_D$, while those with asymptomatic infection develop a sub-patent infection at a rate $r_A$. The sub-patent infection and prophylaxis states then clear infection and return to the susceptible state at rates $r_U$ and $r_P$, respectively. In the susceptible compartment, re-infection can occur, while asymptomatic and sub-patent infections are also susceptible to superinfection, potentially giving rise to further clinical cases. *P. falciparum* parameters are listed in *Appendix 1—table 1*.

The human component of the model is described by the following set of partial differential equations with regard to time $t$ and age $a$:

$$\begin{aligned}
\frac{\partial S}{\partial t} + \frac{\partial S}{\partial a} &= -\Lambda(t - d_E)S + r_U U + r_P P \\
\frac{\partial D}{\partial t} + \frac{\partial D}{\partial a} &= \phi(1 - f_T)\Lambda(t - d_E)(S + A + U) - r_D D \\
\frac{\partial T}{\partial t} + \frac{\partial T}{\partial a} &= \phi f_T \Lambda(t - d_E)(S + A + U) - r_T T \\
\frac{\partial A}{\partial t} + \frac{\partial A}{\partial a} &= (1 - \phi)\Lambda\left(t - d_E\right)(S + U) + r_D D - \phi\Lambda\left(t - d_E\right)A - r_A A \\
\frac{\partial U}{\partial t} + \frac{\partial U}{\partial a} &= r_A A - \Lambda\left(t - d_E\right)U - r_U U \\
\frac{\partial P}{\partial t} + \frac{\partial P}{\partial a} &= r_T T - r_P P
\end{aligned} \tag{3}$$

Note that age- and time-dependence in state variables and parameters, as well as mortality and birth rates, are omitted in equations for clarity.

Accounting for the heterogeneity and age-dependence in mosquito biting rates described above, the force of infection $\Lambda\left(a, t\right)$ and the EIR $\varepsilon\left(a, t\right)$ for age $a$ at time $t$ are given by:

$$\Lambda\left(a, t\right) = \varepsilon\left(a, t\right) b \tag{4}$$

$$\varepsilon\left(a,t\right) = \varepsilon_0\left(t\right)\zeta\psi\left(a\right) \tag{5}$$

Where $\varepsilon_0$ is the mean entomological inoculation rate (EIR) experienced by adults at time $t$, and $b$ is the probability that a human will be infected when bitten by an infectious mosquito.

The mean EIR experienced by adults is represented by:

$$\varepsilon_0\left(t\right) = \frac{\alpha I_M}{\omega} \tag{6}$$

Where $\alpha$ is the mosquito biting rate in humans, $I_M$ is the compartment for adult infectious mosquitoes (see vector model component), and $\omega$ is a normalization constant for the biting rate over various age groups with a population age distribution of $\eta\left(a\right)$, as follows.

$$\omega = \int_0^\infty \eta\left(a\right)\psi\left(a\right)da \tag{7}$$

The probability of infection $b$, probability of clinical symptomatic disease $\phi$, and recovery rate from asymptomatic infection $r_A$, all depend on immunity levels. The acquisition and decay of naturally-acquired immunity is tracked dynamically in the model and is driven by both age and exposure. Naturally-acquired immunity affects three different outcomes in the model, leading to: (1) a reduced probability of developing a blood-stage infection following an infectious bite due to pre-erythrocytic immunity, $I_B$, (2) a reduced probability of progression to clinical disease following infection, dependent on exposure-driven and maternally acquired clinical immunity, $I_{CA}$ and $I_{CM}$, and (3) a reduced probability of a blood-stage infection being detected by microscopy, dependent on acquired immunity to the detectability of infection, $I_D$.

The following partial differential equations represent exposure-driven immunity levels at time $t$ and age $a$.

Pre-erythrocytic immunity:

$$\frac{\partial I_B}{\partial t} + \frac{\partial I_B}{\partial a} = \frac{\varepsilon}{\varepsilon u_B + 1} - \frac{I_B}{d_B} \tag{8}$$

Clinical immunity:

$$\frac{\partial I_{CA}}{\partial t} + \frac{\partial I_{CA}}{\partial a} = \frac{\Lambda}{\Lambda u_C + 1} - \frac{I_{CA}}{d_{CA}} \tag{9}$$

Detection immunity:

$$\frac{\partial I_D}{\partial t} + \frac{\partial I_D}{\partial a} = \frac{\Lambda}{\Lambda u_D + 1} - \frac{I_D}{d_{ID}} \tag{10}$$

Where $u$ parameters represent a refractory period during which the different types of immunity cannot be further boosted after receiving a boost, and where $d$ parameters stand for the mean duration of the different types of immunity.

Maternal immunity is acquired and lost as follows:

$$\frac{\partial I_{CM}}{\partial t} + \frac{\partial I_{CM}}{\partial a} = -\frac{I_{CM}}{d_{CM}} \tag{11}$$
$$I_{CM}\left(t,0\right) = P_{CM}I_{CA}\left(t,20\right)$$

Where $d_{CM}$ is the average duration of maternal immunity, $P_{CM}$ is the proportion of the mother's clinical immunity acquired by the newborn, and $I_{CA}\left(t,20\right)$ denotes the clinical immunity level of a 20-year-old woman.

Immunity levels are converted into time- and age-dependent probabilities using Hill functions.

The probability that a human will be infected when bitten by an infectious mosquito, $b$, can be represented as:

$$b = b_0\left(b_1 + \frac{1 - b_1}{1 + \left(I_B/I_{B0}\right)^{\kappa_B}}\right) \tag{12}$$

Where $b_0$ is the maximum probability of infection (with no immunity), $b_1$ is the maximum relative reduction in the probability of infection due to immunity, and $I_{B0}$ and $\kappa_B$ are scale and shape parameters estimated during model fitting.

The probability of a new blood-stage infection becoming symptomatic, $\phi$, is represented by:

$$\phi = \phi_0 \left( \phi_1 + \frac{1 - \phi_1}{1 + \left( (I_{CA} + I_{CM}) / I_{C0} \right)^{\kappa_C}} \right) \tag{13}$$

Where $\phi_0$ is the maximum probability of becoming symptomatic (with no immunity), $\phi_1$ is the maximum relative reduction in the probability of becoming symptomatic due to immunity, and $I_{C0}$ and $\kappa_C$ are scale and shape parameters, respectively.

Immunity can also lead to blood-stage infections becoming sub-patent with low parasitemias. The probability that an asymptomatic infection is detectable by microscopy, $q$, is represented by:

$$q = d_1 + \frac{1 - d_1}{1 + f_D \left( I_D / I_{D0} \right)^{\kappa_D}} \tag{14}$$

Where $d_1$ is the minimum probability of detectability (with full immunity), and $I_{D0}$ and $\kappa_D$ are scale and shape parameters, respectively. $f_D$ is an age-dependent function modifying the detectability of infection:

$$f_D = 1 - \frac{1 - f_{D0}}{1 + \left( \dfrac{a}{a_D} \right)^{\gamma_D}} \tag{15}$$

With $\gamma_D$ and $a_D$ representing shape and scale parameters, and $f_{D0}$ representing the time-scale at which immunity changes with age.

**Appendix 1—table 1.** *P. falciparum* human model parameter values.
Full details can be found in the original publication (**Griffin et al., 2016**), including references for parameters and intervals for the prior and posterior distributions (median values of the posterior distribution are used in model simulations).

| Parameter | Symbol | Estimate |
|---|---|---|
| **Human infection duration (days)** | | |
| Latent period | $d_E$ | 12 |
| Patent infection | $\frac{1}{r_A}$ | 195 |
| Clinical disease (untreated) | $\frac{1}{r_D}$ | 5 |
| Treatment of clinical disease | $\frac{1}{r_T}$ | 5 |
| Sub-patent infection | $\frac{1}{r_U}$ | 110.299 |
| Prophylaxis | $\frac{1}{r_P}$ | 15 |
| **Age and heterogeneity** | | |
| Age-dependent biting parameter | $\rho$ | 0.85 |
| Age-dependent biting parameter | $a_0$ | 8 years |

*Appendix 1—table 1 Continued on next page*

*Appendix 1—table 1 Continued*

| Parameter | Symbol | Estimate |
|---|---|---|
| Variance of the log heterogeneity in biting rates | $\sigma^2$ | 1.67 |
| **Pre-erythrocytic immunity reducing probability of infection** | | |
| Duration of refractory period in which immunity is not boosted | $u_B$ | 7.19919 days |
| Duration of pre-erythrocytic immunity | $d_B$ | 10 years |
| Maximum probability of infection due to no immunity | $b_0$ | 0.590076 |
| Maximum relative reduction in probability of infection due to immunity | $b_1$ | 0.5 |
| Scale parameter | $I_{B0}$ | 43.8787 |
| Shape parameter | $K_B$ | 2.15506 |
| **Immunity reducing probability of clinical disease** | | |
| Duration of refractory period in which immunity is not boosted | $u_C$ | 6.06349 days |
| Duration of clinical immunity | $d_{CA}$ | 30 years |
| New-born immunity relative to mother's clinical immunity | $P_{CM}$ | 0.774368 |
| Duration of maternal immunity | $d_{CM}$ | 67.6952 days |
| Maximum probability of clinical disease due to no immunity | $\Phi_0$ | 0.791666 |
| Maximum relative reduction in probability of clinical disease due to immunity | $\Phi_1$ | 0.000737 |
| Scale parameter | $I_{C0}$ | 18.02366 |
| Shape parameter | $K_C$ | 2.36949 |
| **Immunity reducing probability of detection** | | |
| Duration of refractory period in which immunity is not boosted | $u_D$ | 9.44512 days |
| Duration of detection immunity | $d_{ID}$ | 10 years |
| Minimum probability of detection due to maximum immunity | $d_1$ | 0.160527 |
| Scale parameter | $I_{D0}$ | 1.577533 |
| Shape parameter | $K_D$ | 0.476614 |
| Scale parameter relating age to immunity | $a_D$ | 21.9 years |
| Time-scale at which immunity changes with age | $f_{D0}$ | 0.007055 |
| Shape parameter relating age to immunity | $\gamma_D$ | 4.8183 |

## *P. vivax* human model component

In the *P. vivax* model, acquisition, and recovery from blood-stage infection in the absence of treatment is also represented by four compartments: susceptible ($S$), untreated symptomatic infection ($I_D$), successfully treated symptomatic infection ($T$), asymptomatic patent blood-stage infection detectable by light microscopy ($I_{LM}$), asymptomatic sub-microscopic infection not detectable by light microscopy, but detectable by PCR ($I_{PCR}$), and prophylaxis ($P$). Additionally, the model represents the liver stage of *P. vivax* infection by tracking average hypnozoite batches in the population. Hypnozoites can form after an infectious bite and remain dormant in the liver for up to several years, which can give rise to relapse blood-stage infections. *P. vivax* parameters are listed in ***Appendix 1—table 2***.

New blood-stage infections can, therefore, originate from either mosquito bites or relapses and are represented by the force of infection $\lambda_H^0$. The force of infection depends on the age-dependent biting rate, the mosquito population size and its level of infectivity, the probability of infection resulting from an infectious bite, the latent period between sporozoite inoculation and development of blood-stage merozoites, $d_E$, as well as relapse infections from the liver stage. Upon

infection, a proportion $\Phi_{LM}$ of humans develop infection detectable by light microscopy (LM), while the remainder have low-density parasitemia and move into the $I_{PCR}$ compartment. A proportion $\Phi_D$ of those with LM-detectable infection develop a clinical episode, of which a proportion $X_T$ are successfully treated with a blood-stage antimalarial. Treated individuals recover from infection at rate $r_T$ and move to the prophylaxis state, which provides temporary protection from reinfection before becoming susceptible again at a rate $r_P$. Recovery from clinical disease to asymptomatic LM-detectable infection, from asymptomatic LM-detectable infection to asymptomatic PCR-detectable infection, and from asymptomatic PCR-detectable infection to susceptibility occur at rates $r_D$, $r_{LM}$, and $r_{PCR}$, respectively. Newborns are susceptible to infection, have no hypnozoites, and are temporarily protected by maternal immunity. Reinfection is possible after recovery, and those with asymptomatic blood stage infections ($I_{LM}$ and $I_{PCR}$) are susceptible to superinfection, potentially giving rise to further clinical cases.

The dynamics of hypnozoite infection in the model describe the accumulation and clearance of $k$ batches of hypnozoites in the liver, whereby each new (super-)infection from an infectious mosquito bite creates a new batch. This process occurs for each model compartment and is described in detail in the original publication (*White et al., 2018*). Hypnozoites from any batch can re-activate and cause a relapse at a rate $kf$, and batches are cleared at a constant rate $k\gamma_L$, which reduces the number of batches from $k$ to $k-1$. For computational efficiency, the possible number of batches in the population must be limited to a maximum value $K$, so that superinfections among the population with $k = K$ do not lead to an increase in hypnozoite batch numbers. We assumed a maximum batch number of 2, which increased computational efficiency and aligned with modeled distributions of hypnozoite batch numbers in the population for the simulated low transmission intensities.

The human component of the model is then described by the following set of partial differential equations with regard to time $t$ and age $a$:

$$
\begin{aligned}
\frac{\partial S^k}{\partial t} + \frac{\partial S^k}{\partial a} =\ & -\lambda_H^0\left(t - d_E\right) S^k - fkS^k + r_{PCR}^k I_{PCR}^k + r_P P^k - \gamma_L k S^k \\
& + \gamma_L\left(k+1\right) S^{k+1} \\[4pt]
\frac{\partial I_{PCR}^k}{\partial t} + \frac{\partial I_{PCR}^k}{\partial a} =\ & -\lambda_H^0\left(t - d_E\right) I_{PCR}^k - fk I_{PCR}^k - r_{PCR}^k I_{PCR}^k + r_{LM} I_{LM}^k \\
& + \lambda_H^0\left(t - d_E\right)\left(1 - \Phi_{LM}^{k-1}\right)(S^{k-1} + I_{PCR}^{k-1}) + fk\left(1 - \Phi_{LM}^k\right)(S^k + I_{PCR}^k) \\
& - \gamma_L k I_{PCR}^k + \gamma_L(k+1) I_{PCR}^{k+1} \\[4pt]
\frac{\partial I_{LM}^k}{\partial t} + \frac{\partial I_{LM}^k}{\partial a} =\ & -\lambda_H^0\left(t - d_E\right) I_{LM}^k - fk I_{LM}^k - r_{LM} I_{LM}^k + r_D I_D^k \\
& + \lambda_H^0\left(t - d_E\right)\left(1 - \Phi_D^{k-1}\right)(\Phi_{LM}^{k-1} S^{k-1} + \Phi_{LM}^{k-1} I_{PCR}^{k-1} + I_{LM}^{k-1}) \\
& + fk\left(1 - \Phi_D^k\right)(\Phi_{LM}^k S^k + \Phi_{LM}^k I_{PCR}^k + I_{LM}^k) - \gamma_L k I_{LM}^k + \gamma_L(k+1) I_{LM}^{k+1} \\[4pt]
\frac{\partial I_D^k}{\partial t} + \frac{\partial I_D^k}{\partial a} =\ & -\lambda_H^0\left(t - d_E\right) I_D^k + \lambda_H^0\left(t - d_E\right) I_D^{k-1} - r_D I_D^k \\
& + \lambda_H^0\left(t - d_E\right) \Phi_D^{k-1}(1 - t)(LMk - 1Sk - 1 + LMk - 1IPCRk - 1 + ILMk - 1) \\
& + fkDk(1 - t)(LMkSk + LMkIPCRk + ILMk) - LkIDk + L(k+1)IDk + 1 \\[4pt]
\frac{\partial T^k}{\partial t} + \frac{\partial T^k}{\partial a} =\ & -\lambda_H^0\left(t - d_E\right) T^k + \lambda_H^0\left(t - d_E\right) T^{k-1} - r_T T^k \\
& + \lambda_H^0\left(t - d_E\right) \Phi_D^{k-1} t(LMk - 1Sk - 1 + LMk - 1IPCRk - 1 + ILMk - 1) \\
& + fkDkt(LMkSk + LMkIPCRk + ILMk) - LkTk + L(k+1)Tk + 1 \\[4pt]
\frac{\partial P^k}{\partial t} + \frac{\partial P^k}{\partial a} =\ & -\lambda_H^0\left(t - d_E\right) P^k + \lambda_H^0\left(t - d_E\right) P^{k-1} + r_T T^k - r_P P^k - \gamma_L k P^k \\
& + \gamma_L\left(k+1\right) P^{k+1}
\end{aligned}
\tag{16}
$$

Where $fk$ and $\gamma_L k$ are the relapse and clearance rates of hypnozoite batch $k$, respectively. Age- and time-dependence in state variables and parameters, as well as mortality and birth rates, are omitted in the equations for clarity.

The equations reflect the accumulation of hypnozoite batches from $k$ to $k+1$ due to infections arising from new infectious bites ($\lambda_H^0$), but not due to relapse infections ($fk$). The total force of blood-stage infection is, therefore:

$$
\lambda_H^k = \left(t - d_E\right) = \lambda_H^0\left(t - d_E\right) + kf
\tag{17}
$$

Similar to the *P. falciparum* model, the force of infection from mosquito bites accounts for heterogeneity and age-dependence in mosquito biting rates as follows:

$$\lambda_H^0(a, t) = \varepsilon(a, t) b \tag{18}$$

$$\varepsilon(a, t) = \varepsilon_0(t) \zeta \psi(a) \tag{19}$$

$$\varepsilon_0(t) = \frac{\alpha I_M}{\omega} \tag{20}$$

$$\omega = \int_0^\infty \eta(a) \psi(a) \, da \tag{21}$$

Where $\varepsilon_0$ is the mean entomological inoculation rate (EIR) experienced by adults at time $t$, and $b$ is the probability that a human will be infected when bitten by an infectious mosquito. In the *P. vivax* model, $b$ is a constant and does not depend on immunity levels. In the calculation of the mean EIR experienced by adults, $\alpha$ is the mosquito biting rate in humans, $I_M$ is the compartment for adult infectious mosquitoes (see vector model component), and $\omega$ is a normalization constant for the biting rate over various age groups with a population age distribution of $\eta(a)$.

Transmission dynamics in the model are influenced by anti-parasite ($A_P$) and clinical immunity ($A_C$) against *P. vivax*. Anti-parasite immunity is assumed to reduce the probability of blood-stage infections achieving high enough density to be detectable by light microscopy ($\Phi_{LM}$) and to increase the rate at which sub-microscopic infections are cleared ($r_{PCR}$). Clinical immunity reduces the probability that LM-detectable infections progress to clinical disease ($\Phi_D$). Like for *P. falciparum*, the dynamics of the acquisition and decay of naturally-acquired immunity in the model depend on age and exposure. For *P. vivax*, immunity levels are boosted by both primary infections and relapses and are described by the following set of partial differential equations with regards to time $t$ and age $a$:

Anti-parasite immunity:

$$
\begin{aligned}
\frac{\partial A_P^0}{\partial t} + \frac{\partial A_P^0}{\partial a} &= -\lambda_H^0(t - d_E) A_P^0 - r_{par} A_P^0 + \gamma_L A_P^1 \\
\frac{\partial A_P^k}{\partial t} + \frac{\partial A_P^k}{\partial a} &= \frac{\lambda_H^k(t - d_E)}{\lambda_H^k(t - d_E) u_{par} + 1} - \lambda_H^0(t - d_E) A_P^k \\
&\quad + \lambda_H^0(t - d_E) A_P^{k-1} - r_{par} A_P^k - \gamma_L k A_P^k + \gamma_L(k+1) A_P^{k+1} \\
\frac{\partial A_P^K}{\partial t} + \frac{\partial A_P^K}{\partial a} &= \frac{\lambda_H^K(t - d_E)}{\lambda_H^K(t - d_E) u_{par} + 1} + \lambda_H^0(t - d_E) A_P^{K-1} - r_{par} A_P^K - \gamma_L K A_P^K
\end{aligned}
\tag{22}
$$

Clinical immunity:

$$
\begin{aligned}
\frac{\partial A_C^0}{\partial t} + \frac{\partial A_C^0}{\partial a} &= -\lambda_H^0(t - d_E) A_C^0 - r_C A_C^0 + \gamma_L A_C^1 \\
\frac{\partial A_C^k}{\partial t} + \frac{\partial A_C^k}{\partial a} &= \frac{\lambda_H^k(t - d_E)}{\lambda_H^k(t - d_E) u_C + 1} - \lambda_H^0(t - d_E) A_C^k \\
&\quad + \lambda_H^0(t - d_E) A_C^{k-1} - r_C A_C^k - \gamma_L k A_C^k + \gamma_L(k+1) A_C^{k+1} \\
\frac{\partial A_C^K}{\partial t} + \frac{\partial A_C^K}{\partial a} &= \frac{\lambda_H^K(t - d_E)}{\lambda_H^K(t - d_E) u_C + 1} + \lambda_H^0(t - d_E) A_C^{K-1} - r_C A_C^K - \gamma_L K A_C^K
\end{aligned}
\tag{23}
$$

Where $u$ parameters represent a refractory period during which the different types of immunity cannot be further boosted after receiving a boost, and where $r$ parameters stand for the rates of decay of the different types of immunity. $k$ refers to the hypnozoite batch (with $K$ being the maximum number of hypnozoite batches).

The levels of maternally acquired anti-parasite and clinical immunity are calculated as:

$$A_{P,\,mat}(t,\,a) = P_{mat} A_P^*(t - a, 20) e^{-\frac{a}{d_{mat}}} \tag{24}$$

$$A_{C,\,mat}\,(t,\,a) = P_{mat}A_C^*\,(t-a,20)\,e^{-\frac{a}{d_{mat}}}$$

(25)

Where $d_{mat}$ is the average duration of maternal immunity, $P_{mat}$ is the proportion of the mother's immunity acquired by the newborn, and $A_P^*\,(t-a,20)$ and $A_C^*\,(t-a,20)$ denote the anti-parasite and clinical immunity levels of a 20-year-old woman averaged over their hypnozoite batches, respectively.

Immunity levels are then converted into time-dependent probabilities using Hill functions.

The probability that a blood-stage infection becomes detectable by LM, $\Phi_{LM}$, can be represented as:

$$\Phi_{LM} = \Phi_{LM,min} + \left(\Phi_{LM,max} - \Phi_{LM,min}\right)\frac{1}{1+\left(\frac{A_P^k + A_{P,mat}}{A_{LM,50\%}}\right)^{K_{LM}}}$$

(26)

Where $\Phi_{LM,min}$ is the minimum probability of LM-detectable infection (with full immunity), $\Phi_{LM,max}$ is the maximum probability of LM-detectable infection (with no immunity), and $A_{LM,50\%}$ and $K_{LM}$ are scale and shape parameters estimated during model fitting.

The probability of an LM-detectable blood-stage infection becoming symptomatic, $\Phi_D$, is represented by:

$$\Phi_D = \Phi_{D,min} + \left(\Phi_{D,max} - \Phi_{D,min}\right)\frac{1}{1+\left(\frac{A_C^k + A_{C,mat}}{A_{D,50\%}}\right)^{K_D}}$$

(27)

Where $\Phi_{D,min}$ is the minimum probability of developing a clinical episode (with full immunity), $\Phi_{D,max}$ is the maximum probability of a clinical episode (with no immunity), and $A_{D,50\%}$ and $K_D$ are scale and shape parameters.

The recovery rate from $I_{PCR}$ is calculated as $\frac{1}{d_{PCR}^k}$. The average duration of a low-density blood-stage infection, $d_{PCR}^k$, is represented by:

$$d_{PCR}^k = d_{PCR,min} + \left(d_{PCR,max} - d_{PCR,min}\right)\frac{1}{1+\left(\frac{A_P^k + A_{P,mat}}{A_{PCR,50\%}}\right)^{K_{PCR}}}$$

(28)

Where $d_{PCR,min}$ is the minimum duration (with full immunity), $d_{PCR,max}$ is the maximum duration (with no immunity), and $A_{PCR,50\%}$ and $K_{PCR}$ are scale and shape parameters.

**Appendix 1—table 2.** *P. vivax* human model parameter values.
Full details can be found in the original publication (*White et al., 2018*) including references for parameters and intervals for the prior and posterior distributions.

| Parameter | Symbol | Estimate |
|---|---|---|
| Human infection duration (days) | | |
| Latent period | $d_E$ | 10 |
| | $\frac{1}{r_{LM}}$ | 10 |
| Light microscopy-detectable asymptomatic infection | | |
| | $\frac{1}{r_D}$ | 5 |
| Clinical disease (untreated) | | |

*Appendix 1—table 2 Continued on next page*

*Appendix 1—table 2 Continued*

| Parameter | Symbol | Estimate |
|---|---|---|
| Treatment of clinical disease | $\frac{1}{r_T}$ | 1 |
| Prophylaxis | $\frac{1}{r_P}$ | 28 |
| **Age, heterogeneity, and probability of infection** | | |
| Age-dependent biting parameter | $\rho$ | 0.85 |
| Age-dependent biting parameter | $a_0$ | 8 years |
| Variance of the log heterogeneity in biting rates | $\sigma^2$ | 1.29 |
| Probability of blood-stage infection upon infectious mosquito bite | $b$ | 0.5 |
| **Hypnozoite parameters** | | |
| Relapse rate | $f$ | 0.024 per day |
| Clearance rate | $\gamma_L$ | 0.0026 per day |
| **Maternal immunity** | | |
| New-born immunity relative to mother's clinical immunity | $P_{mat}$ | 0.421 |
| Duration of maternal immunity | $d_{mat}$ | 35.148 days |
| **Anti-parasite immunity reducing probability of light microscopy-detectable infection and duration of PCR-detectable infection** | | |
| Duration of refractory period in which immunity is not boosted | $u_{par}$ | 19.77 days |
| Duration of anti-parasite immunity | $\frac{1}{r_{par}}$ | 10 years |
| Maximum probability of detectability by light microscopy due to no immunity | $\Phi_{LM,max}$ | 0.8918 |
| Minimum probability of detectability by light microscopy due to full immunity | $\Phi_{LM,min}$ | 0.0043 |
| Scale parameter for detectability by light microscopy | $A_{LM,50\%}$ | 27.52 |
| Shape parameter for detectability by light microscopy | $K_{LM}$ | 2.403 |
| Maximum duration of PCR-detectable infection due to no immunity | $d_{PCR,max}$ | 70 days |
| Minimum duration of PCR-detectable infection due to full immunity | $d_{PCR,min}$ | 10 days |
| Scale parameter for duration of PCR-detectable infection | $A_{PCR,50\%}$ | 9.9 |
| Shape parameter for duration of PCR-detectable infection | $K_{PCR}$ | 4.602 |
| **Clinical immunity reducing probability of clinical disease** | | |
| Duration of refractory period in which immunity is not boosted | $u_C$ | 7.85 days |
| Duration of detection immunity | $\frac{1}{r_C}$ | 30 years |
| Maximum probability of clinical disease due to no immunity | $\Phi_{D,\,max}$ | 0.8605 |
| Minimum probability of clinical disease due to full immunity | $\Phi_{D,\,min}$ | 0.018 |
| Scale parameter for clinical disease | $A_{D,50\%}$ | 11.538 |
| Shape parameter for clinical disease | $K_D$ | 2.250 |

## Mosquito component of the *P. falciparum* and *P. vivax* model

The mosquito components of the *P. falciparum* and *P. vivax* models capture adult mosquito transmission dynamics, as well as larval population dynamics, and are nearly identical. Modeled vector bionomics correspond to *Anopheles gambiae s.s.* and *Anopheles punctulatus* for *P. falciparum* and *P. vivax* transmission, respectively.

### Mosquito transmission model

Adult mosquitoes move between three states, $S_M$ (susceptible), $E_M$ (exposed), and $I_M$ (infectious), as follows:

$$\frac{dS_M}{dt} = \Lambda_M S_M + \beta(t) - \mu S_M$$
$$\frac{dI_M}{dt} = \Lambda_M S_M - \Lambda_M (t - \tau_M) S_M (t - \tau_M) P_M - \mu E_M \tag{29}$$
$$\frac{dI_M}{dt} = \Lambda_M (t - \tau_M) S_M (t - \tau_M) P_M - \mu I_M$$

is the force of infection from humans to mosquitos, $\beta(t)$ represents the time-varying adult mosquito emergence rate, $\mu$ is the adult mosquito death rate, and $\tau_M$ represents the extrinsic incubation period. $P_M$ represents the probability that a mosquito survives between being infected and sporozoites appearing in the salivary glands and is calculated as $\exp(-\mu \tau_M)$.

The force of infection experienced by the vector is the sum of the contribution to mosquito infections from all human infectious states. As described for the human model components for both species, it also depends on the mosquito biting rate in humans (which depends on net usage), $\alpha$, and a normalization constant for the biting rate over various age groups, $\omega$.

### Force of infection experienced by mosquitoes in the *P. falciparum* model

In the *P. falciparum* model, the force of infection acting on mosquitoes is represented by:

$$\Lambda_M (t) = \frac{\alpha}{\omega} \iint_{\zeta \alpha} \zeta \psi (a) \left( c_D D (\zeta, a, t - \tau_1) + c_T T (\zeta, a, t - \tau_1) \right.$$
$$\left. + c_A A (\zeta, a, t - \tau_1) + c_U U (\zeta, a, t - \tau_1) \, da \, d\zeta \right. \tag{30}$$

Where $c_D$, $c_T$, $c_A$, and $c_U$ represent the human-to-mosquito infectiousness for untreated symptomatic infection, treated symptomatic infection, asymptomatic infection, and asymptomatic sub-patent infection, respectively. $\tau_1$ is the time lag between parasitemia with asexual parasite stages and gametocytemia to account for the time to *P. falciparum* gametocyte development.

The infectiousness of humans with asymptomatic infection, $c_A$, is reduced by a lower probability of detection of infection by microscopy due to the assumption that lower parasite densities are less detectable. While infectiousness parameters $c_D$ and $c_U$ are constant, infectivity for asymptomatic infection is calculated as follows:

$$c_A = c_U + (c_D - c_U) q^{\gamma_1} \tag{31}$$

Where $q$ is the immunity-dependent probability that an asymptomatic infection is detectable by microscopy (**Equation 14**) and the parameter $\gamma_1$ was estimated during the original model fitting in previous publications (**Griffin et al., 2010**; **Griffin et al., 2014**; **Griffin et al., 2016**).

### Force of infection experienced by mosquitoes in the *P. vivax* model

In the *P. vivax* model, the force of infection acting on mosquitoes is represented by:

$$\Lambda_M (t) = \frac{\alpha}{\omega} \iint_{\zeta \alpha} \zeta \psi (a) \left( c_D I_D (\zeta, a, t) + c_T T (\zeta, a, t) + c_{LM} I_{LM} (\zeta, a, t) \right.$$
$$\left. + c_{PCR} I_{PCR} (\zeta, a, t) \, da \, d\zeta \right. \tag{32}$$

Where $c_D$, $c_T$, $c_{LM}$, and $c_{PCR}$ represent the human-to-mosquito infectiousness for untreated symptomatic infection, treated symptomatic infection, asymptomatic LM-detectable infection and asymptomatic PCR-detectable infection, respectively. Due to the quicker development of *P. vivax* gametocytes compared to *P. falciparum*, there is assumed to be no delay between infection and infectiousness in humans.

## Larval development

For both *P. falciparum* and *P. vivax* the larval stage model, shown in the following equations, is based on the previously described model in *White et al., 2011*. Female adult mosquitoes lay eggs at a rate $\beta_L$ . Upon hatching from eggs, larvae progress through early and late larvae stages ($E$ and $L$ compartments) before developing into to the pupal stage $P_L$ . Adult female mosquitoes emerge from the pupal stage in *Equation (29)*, which is calculated as $\beta = 0.5 \frac{P_L}{d_P}$.

$$\frac{dE}{dt} = \beta_L(S_M + E_M + I_M) - \mu_E \left(1 + \frac{E + L}{K}\right) E - \frac{E}{d_E}$$
$$\frac{dL}{dt} = \frac{E}{d_E} - \mu_L \left(1 + \gamma \frac{E + L}{K}\right) L - \frac{L}{d_L} \tag{33}$$
$$\frac{dP_L}{dt} = \frac{L}{d_L} - \mu_P P_L - \frac{P_L}{d_P}$$

The duration of each larval stage is represented by $d_E$ , $d_L$, and $d_P$. The larval stages are regulated by density-dependent mortality rates, with a time-varying carrying capacity, $K$, that represents the ability of the environment to sustain breeding sites through different periods of the year and with the density of larvae in relation to the carrying capacity regulated by a parameter $\gamma$. Since seasonality in transmission dynamics was not modeled at the country level in this analysis, the carrying capacity was assumed to be constant throughout the year. The carrying capacity determines the mosquito density and hence the baseline transmission intensity in the absence of interventions. It is calculated as:

$$K = M_0 \frac{2 d_L \mu_0 \left(1 + d_P \mu_P\right) \gamma \left(\lambda_M + 1\right)}{\left(\frac{\lambda_M}{\mu_L d_E} - \frac{1}{\mu_L d_L} - 1\right)} \tag{34}$$

Where $M_0$ is the initial female mosquito density, $\mu_0$ is the baseline mosquito death rate, and $\lambda_M$ is defined as:

$$\lambda_M = \begin{aligned} &-0.5 \left(\gamma \frac{\mu_L}{\mu_E} - \frac{d_E}{d_L} + (\gamma - 1) \mu_L d_E\right) \\ &+ \sqrt{0.25 \left(\gamma \frac{\mu_L}{\mu_E} - \frac{d_E}{d_L} + (\gamma - 1) \mu_L d_E\right)^2 + \gamma \frac{\beta_L \mu_L d_E}{2 \mu_E \mu_0 d_L \left(1 + d_P \mu_P\right)}} \end{aligned} \tag{35}$$

In this equation, the number of eggs laid per day, $\beta_L$ , is defined as:

$$\beta_L = \frac{\beta_{L_{max}} \mu e^{-\frac{\mu}{f_R}}}{\mu \left(e^{\frac{\mu}{f_R}} - 1\right) \left(1 - e^{-\frac{\mu}{f_R}}\right)} \tag{36}$$

Where $\beta_{L_{max}}$ is the maximum number of eggs per oviposition per mosquito. The adult mosquito death rate μ and the mosquito feeding rate $f_R$ are affected by the use of ITNs and further described in the following section on modeling vector control. Full details on the derivation of the egg-laying rate $\beta_L$ and the carrying capacity $K$ have been previously published (*White et al., 2011*).

## Modeling the impact of ITNs

ITNs are modeled as described previously (*Griffin et al., 2010*; *Griffin et al., 2016*). Mosquito population and transmission dynamics are affected by the use of ITNs in four ways: the mosquito death rate is increased, the feeding or gonotrophic cycle is increased, the proportion of bites taken on protected and unprotected people is changed, and the proportion of bites taken on humans relative to animals is affected. The probability that a blood-seeking mosquito successfully feeds on a human (as opposed to being repelled or killed) will depend on species-dependent bionomics and behaviors of the mosquito, as well as the anti-vectorial interventions present in the human population. Parameter values can be found in *Appendix 1—table 3*.

## Mosquito feeding behavior

In the model there are four possible outcomes of a mosquito feeding attempt:

1. The mosquito bites a non-human host.
2. The mosquito attempts to bite a human host but is killed by the ITN before biting.
3. The mosquito successfully feeds on a human host and survives that feeding attempt.
4. The mosquito attempts to bite a human host but is repelled by the ITN without feeding, and repeats the attempt to find a blood meal source.

We define the probability of a mosquito biting a human host during a single attempt as $y_i$, the probability that a mosquito bites a human host and survives the feeding attempt as $w_i$, and the probability of a mosquito being repelled without feeding as $z_i$. These probabilities exclude natural vector mortality, so that for a population without protection from ITNs (e.g. prior to their introduction), $y_1 = w_1 = 1$ and $z_1 = 0$.

The presence of ITNs modifies these probabilities of surviving a feeding attempt or being repelled without feeding. Upon entering a house with ITNs, mosquitoes can experience three different outcomes: being repelled by the ITN without feeding (probability $r_N$), being killed by the ITN before biting (probability $d_N$), or feeding successfully (probability $s_N$). It is assumed that all biting attempts inside a house occur in humans. The repellency of ITNs in terms of the insecticide and barrier effect decays over time, giving the following probabilities:

$$r_N = (r_{N0} - r_{NM}) \exp(-t\gamma_N) + r_{NM} \tag{37}$$

$$d_N = d_{N0} \exp(-t\gamma_N) \tag{38}$$

$$s_N = 1 - r_N - d_N \tag{39}$$

Where $r_{N0}$ is the maximum probability of a mosquito being repelled by a bednet and $r_{NM}$ is the minimum probability of being repelled by a bednet that no longer has insecticidal activity and possibly holes reducing the barrier effect. $\gamma_N$ represents the rate of decay of the effect of ITNs over time $t$ since their distribution and is calculated as $\frac{\log(2)}{LLIN\ half-life}$. The killing effect of ITNs decreases at the same constant rate from a maximum probability of $d_{N0}$. In model simulations, ITNs are distributed every three years.

With $i = 1$ representing the population not covered by an ITN and $i = 2$ representing the population covered by an ITN, this gives the following probabilities of successfully feeding, $W$, and being repelled without feeding, $Z$, during a single feeding attempt on a human:

$$W = \sum_{i=1}^2 w_i c_i \qquad w = \begin{cases} 1 & if\ i = 1 \\ 1 - \Phi_b + \Phi_b s_N & if\ i = 2 \end{cases} \tag{40}$$

$$Z = \sum_{i=1}^2 z_i c_i \qquad z = \begin{cases} 0 & if\ i = 1 \\ \Phi_b r_N & if\ i = 2 \end{cases} \tag{41}$$

Where $c_i$ is the proportion of the population in the respective group, and $\Phi_b$ is the proportion of bites taken on humans in bed, which was derived from previous publications (*Griffin et al., 2010*).

During a single feeding attempt (which may be on animals or humans), the average probability of mosquitoes feeding or being repelled without feeding, $\overline{W}$ and $\overline{Z}$, are then:

$$\overline{W} = 1 - Q_0 + Q_0 W \tag{42}$$

$$\overline{Z} = Q_0 Z \tag{43}$$

Where $Q_0$ is the proportion of bites taken on humans in the absence of any vector control intervention.

## Effect of ITNs on mosquito mortality

The average probability of mosquitoes being repelled without feeding in the model affects the mosquito feeding rate, $f_R$, as follows:

$$f_R = \frac{1}{\frac{\delta_1}{\left(1 - \overline{Z}\right)} + \delta_2}$$

(44)

Where $\delta_1$ is the time spent looking for a blood meal in the absence of vector control, and $\delta_2$ is the time spent resting between blood meals, which is assumed to be unaffected by ITN usage.

The average probabilities of feeding or being repelled also affect the probability of surviving the period of feeding, $p_1$, as follows:

$$p_1 = \frac{\overline{W} \exp\left(-\mu_0 \delta_1\right)}{1 - \overline{Z} \exp\left(-\mu_0 \delta_1\right)}$$

(45)

Where $\mu_0$ is the baseline mosquito death rate in the absence of interventions.

The probability of surviving the period of resting, $p_2$, is not affected by ITNs:

$$p_2 = \exp\left(-\mu_0 \delta_2\right)$$

(46)

This allows to calculate the mosquito mortality rate affecting mosquito population dynamics in the set of **Equations (29)**:

$$\mu = -f_R \ln\left(p1 * p2\right)$$

(47)

## Effect of ITNs on the force of infection acting on humans and mosquitoes

In the presence of ITNs, the anthropophagy (the proportion of successful bites which are on humans) of mosquitoes is represented by parameter $Q$. This is affected by ITN usage as follows:

$$Q = 1 - \frac{1 - Q_0}{\overline{W}}$$

(48)

Further details on the assumptions in this calculation can be found in an earlier publication (**Griffin et al., 2010**).

This then gives the biting rate on humans, $\alpha$, as shown in the equations for the force of infection experienced by humans (**Equations 4–6**) and by mosquitoes (**Equations 4–6**):

$$\alpha = Q f_R \frac{w_{int}}{W}$$

(49)

## Effect of ITNs on larval development

The mosquito death rate μ and the feeding rate $f_R$ also influence the calculation of the carrying capacity $K$ and the egg-laying rate $\beta_L$ in **Equation 34** and **Equation 36**, thereby affecting larval development.

**Appendix 1—table 3.** Mosquito model and insecticide-treated net (ITN) parameter.
Full details on parameter values can be found in the original publications (**Griffin et al., 2010**; **Griffin et al., 2016**; **White et al., 2011**; **White et al., 2018**), including references and intervals for the prior and posterior distributions for fitted parameters (median values of the posterior distribution are used in model simulations).

|  | *P. falciparum (Anopheles gambiae s.s.)* | *P. vivax (Anopheles punctulatus)* |
| --- | --- | --- |
| Infectiousness of humans to mosquitoes | | |

*Appendix 1—table 3 Continued on next page*

*Appendix 1—table 3 Continued*

| | | P. falciparum (Anopheles gambiae s.s.) | P. vivax (Anopheles punctulatus) |
|---|---|---|---|
| Lag from parasites to infectious gametocytes | $\tau_1$ | 12.5 days | - |
| Untreated clinical disease | $c_D$ | 0.068 | 0.8 |
| Treated clinical disease | $c_T$ | 0.022 | 0.4 |
| Sub-patent infection | $c_U$ | 0.0062 | - |
| Parameter for infectiousness of asymptomatic infection | $\gamma_1$ | 1.82425 | - |
| Light microscopy-detectable infection | $c_{LM}$ | - | 0.1 |
| PCR-detectable infection | $c_{PCR}$ | - | 0.035 |
| **Mosquito population model** | | | |
| Daily mortality of adult mosquitoes with no interventions | $\mu_0$ | 0.132 | 0.167 |
| Extrinsic incubation period | $\tau_M$ | 10 days | 8.4 days |
| **Larval model** | | | |
| Early instar larval developmental period | $d_E$ | 6.64 days | 6.64 days |
| Late instar developmental period | $d_L$ | 3.72 days | 3.72 days |
| Pupal developmental period | $d_P$ | 0.643 days | 0.643 days |
| Daily mortality rate of early-stage larvae (density-dependent) | $\mu_E$ | 0.0338 | 0.0338 |
| Daily mortality rate of late-stage larvae (density-dependent) | $\mu_L$ | 0.0348 | 0.0348 |
| Daily mortality rate of pupae (density-independent) | $\mu_P$ | 0.249 | 0.249 |
| Effect of density dependence on late instars relative to early instars | $\gamma$ | 13.25 | 13.25 |
| Maximum number of eggs per oviposition per mosquito | $\beta_{L_{max}}$ | 21.2 | 21.2 |
| **Mosquito behavior** | | | |
| Mean duration of host-seeking in the absence of vector control interventions | $\delta_1$ | 0.69 days | 0.69 days |
| Mean duration of resting between blood meals | $\delta_2$ | 2.31 days | 2.31 days |
| Proportion of bites taken on humans (anthropophagy) in the absence of vector control interventions | $Q_0$ | 0.92 | 0.5 |
| Proportion of bites taken on humans indoors and in bed | $\Phi_b$ | 0.89 | 0.9 |
| **Effect of ITNs** | | | |
| Maximum probability of a mosquito being repelled by a ITN with full insecticidal and barrier effect | $r_{N0}$ | 0.56 | 0.6 |
| Minimum probability of a mosquito being repelled by a ITN after decay | $r_{NM}$ | 0.24 | 0.2 |
| ITN half-life | - | 2.64 years | 2.64 years |
| Maximum probability of a mosquito being killed by a ITN with full insecticidal and barrier effect | $d_{N0}$ | 0.41 | 0.3 |

## Assumptions in model outcomes

### Model dynamics over time

To represent long-term reductions in clinical burden, model simulations were run until a new equilibrium was reached post-intervention for all ITN usage levels, which corresponded to 75 years for *P. falciparum* and 175 years for *P. vivax*. As shown in **Appendix 1—figure 2**, when ITNs are

continuously distributed over time, clinical incidence outcomes initially fluctuate before reaching a long-term equilibrium due to various effects on population immunity and mosquito population dynamics in the model. For example, in high-transmission *P. falciparum* settings, clinical incidence experiences a steep initial decline after ITN introduction, before gradually rebounding to an equilibrium value (*Appendix 1—figure 2A*). In the *P. vivax* model, stabilization at an equilibrium transmission level was further delayed due to the presence of hypnozoites in a deterministic framework, whereby even an extremely small reservoir could lead to rebounds in clinical infections after decades. To limit *P. vivax* simulations to a computationally feasible time period, we prevented this rebound by introducing the assumption that once a hypnozoite prevalence of less than 1 in 1,00,000 is reached in the population, the reservoir is further depleted and cannot lead to a renewed chain of transmission (*Appendix 1—figure 2B*).

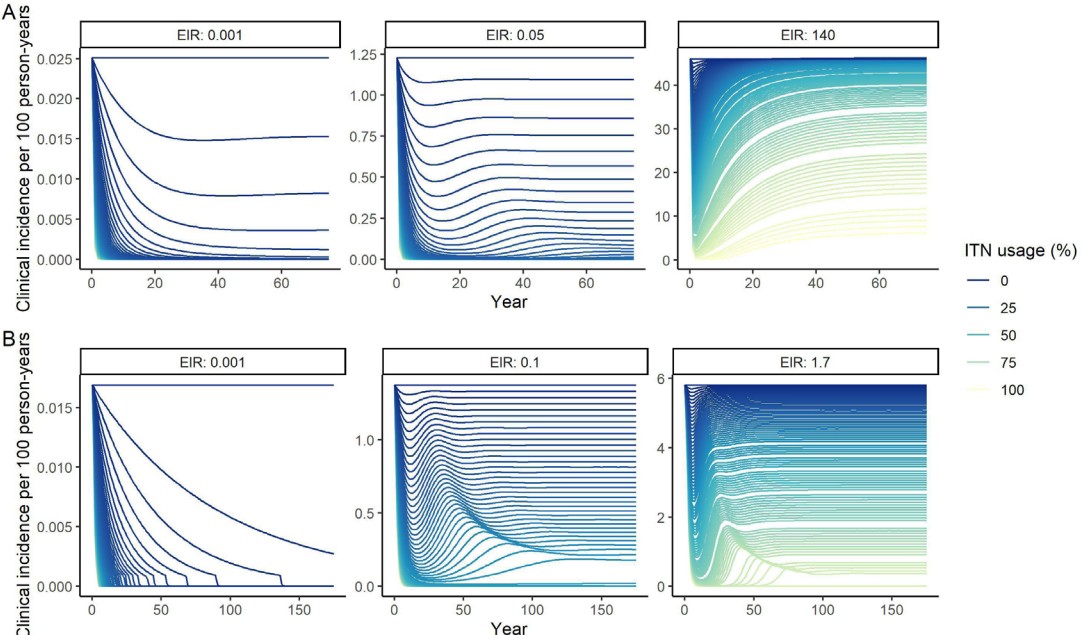

**Appendix 1—figure 2.** Modeled impact of insecticide-treated net (ITN) usage on clinical incidence over time for three representative entomological inoculation rates (EIR) for (**A**) *P. falciparum* and (**B**) *P. vivax*. EIRs represent the minimum, median, and maximum EIRs of the global population distribution. Lines represent increments of 1% of ITN usage.

## Clinical incidence and assumptions about case detection

We simulated clinical incidence assuming cases would be detected through weekly active case detection (ACD). ACD represents a more sensitive method to assess disease burden and was used in the majority of studies used to calibrate *P. falciparum* and *P. vivax* models (*Griffin et al., 2010*; *White et al., 2018*). This assumption results in higher case incidence than reported case numbers because not everyone seeks care at a health clinic for a clinical episode (*Griffin et al., 2010*). As estimated in previous publications, weekly ACD was assumed to detect 72.3% and 13.4% of all *P. falciparum* and *P. vivax* clinical cases detected by daily ACD, respectively (*Griffin et al., 2010*; *White et al., 2018*; *Battle et al., 2015*).

## Country-level data and modeling assumptions on the global malaria distribution

To represent the global distribution of malaria, a *P. falciparum* prevalence in 2–10 year-olds ($PfPR_{2-10}$) (2000) raster layer (*Weiss et al., 2019*) was clipped to a *P. falciparum* transmission spatial limits (2010) raster layer (*Gething et al., 2011*) obtained from the Malaria Atlas Project. Country shapefiles, obtained from geoBoundaries (*Runfola et al., 2020*), were overlaid on prevalence estimates, and the mean $PfPR_{2-10}$ within each boundary was calculated. A similar process was completed for *P. vivax* using $PvPR_{0-99}$ (2000) and *P. vivax* transmission spatial limits (2010) raster layers (*Battle et al., 2019*). WorldPop gridded 2000 global population estimates (*Tatem, 2017*) were summed within

boundaries to output the total population at risk of malaria infection living within each country. For both species, parasite prevalence was then matched to modeled EIR associated with the closest prevalence estimate. The group of countries with the lowest transmission intensity included those with an EIR of 0.001 or lower.

In our analysis, we assumed that most of sub-Saharan Africa was not endemic for *P. vivax*, because *P. vivax* prevalence and incidence could not be estimated (*Battle et al., 2019*). Even though there is evidence for low-level *P. vivax* endemicity throughout the continent, there is no routine surveillance for non-*P. falciparum* cases and the prevalence of the Duffy-negative phenotype among African populations is protective against endemic transmission of *P. vivax* (*Battle et al., 2019*). Therefore, our estimates for the population at risk of *P. vivax* malaria do not include much of sub-Saharan Africa (except the Horn of Africa).

Although model simulations were matched to country-level prevalence, we did not aim to capture the wide geographic variation in malaria epidemiology in detail. For example, in all simulations with the *P. vivax* model, we fixed the relapse and hypnozoite clearance rates, based on the original parameter values used in the calibrated model in Papua New Guinea (*White et al., 2018*). The timings of relapse are thought to follow different patterns in different geographical areas, with a particular distinction between tropical strains relapsing quickly after initial infection and temperate strains relapsing only after 6–12 months (*Battle et al., 2014*). Nevertheless, projections from the model calibrated to sub-national Papua New Guinean data were also shown to be in line with global epidemiological patterns at various prevalence levels (*White et al., 2018*). Similarly, we did not account for the geographic variation in dominant malaria vector species, which are particularly diverse across *P. vivax* endemic areas (*Sinka et al., 2012*).

In all model simulations and analyses, we assumed infections with the two parasite species to be independent, in line with the presentation of estimates from Malaria Atlas Project. Therefore, in each setting, we considered total malaria cases to represent the sum of modeled *P. falciparum* and *P. vivax* cases, total malaria prevalence to represent the sum of *P. falciparum* and *P. vivax* parasite prevalence, and the total EIR to represent the sum of average *P. falciparum*- and *P. vivax*-infectious bites per person year. With the geographical areas endemic for the two species overlapping in many locations, we assumed the population at risk of malaria in each setting to represent the higher of the population at risk of *P. falciparum* or of *P. vivax*.

## Relationship between distribution and usage of ITNs

As described in the manuscript, the non-linear relationship between costs and ITN usage was accounted for by converting the modeled population usage into the required number of ITNs to be distributed to achieve this usage. For this, a published methodology was used; full assumptions and definitions can be found in the original publication (*Bertozzi-Villa et al., 2021*). Equations are detailed below and parameter values for the application in this paper are summarized in *Appendix 1—table 4*.

First, the simulated ITN usage was converted into ITN population access based on observed ITN use rates. By definition:

$$ITN\ access = \frac{ITN\ usage}{ITN\ use\ rate}$$

Since access in the population cannot exceed 1, the modeled ITN usage could not be higher than the assumed use rate.

Second, a Loess curve was fitted to 2020 data on net access and nets per capita per country-month from across Africa, reproducing a similar relationship as shown in the original publication (*Appendix 1—figure 3*, *Bertozzi-Villa et al., 2021*). The net access derived for a given usage was then converted into nets per capita using the Loess curve. We extrapolated the trend for higher access levels and assumed that all access levels below the minimum observed would require the same nets per capita (i.e. the same cost) to achieve.

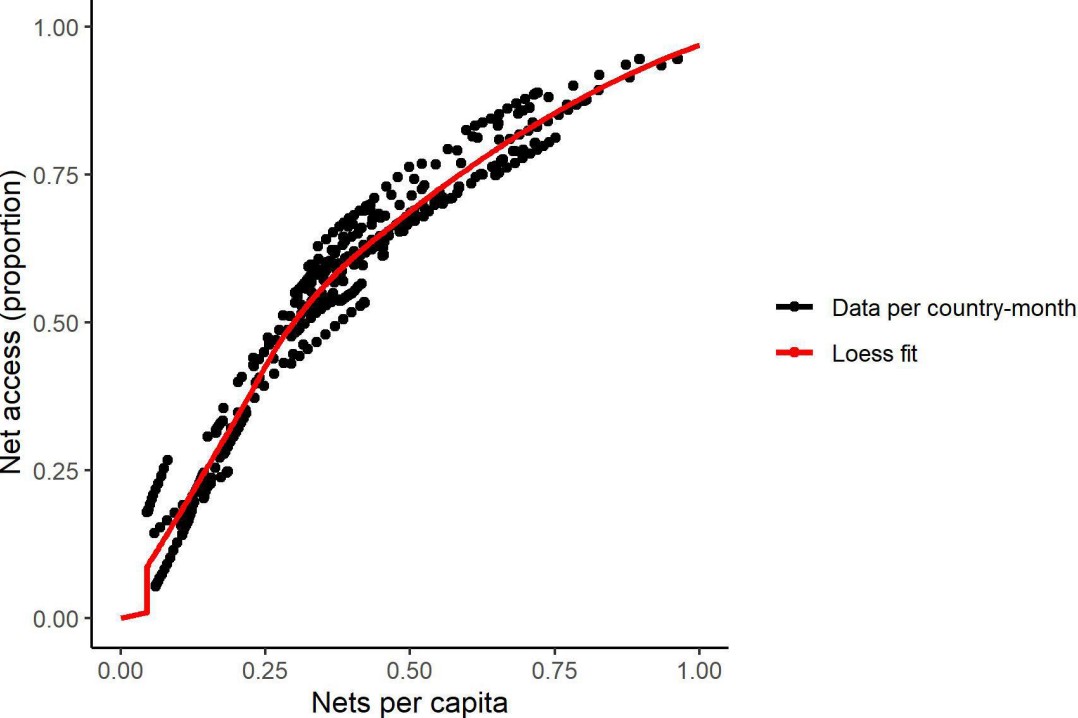

**Appendix 1—figure 3.** Relationship between access and nets per capita in 2020 (generated from data in *Bertozzi-Villa et al., 2021*).

Lastly, the nets per capita were converted into the nets distributed per person-year, accounting for net retention over time and assuming a distribution frequency of every 3 years. Like in the original publication, ITNs were assumed to be lost from the population following a smooth compact function after distribution, so that the proportion of nets retained over time, $p\left(t\right)$, equals:

$$p(t) = \begin{cases} e^{\kappa - \dfrac{\kappa}{1-\left(\frac{t}{\tau}\right)^2}} & if\ t < \tau \\ 0 & if\ t \geq \tau \end{cases}$$

Where $\kappa$ is a fitted rate parameter estimated from the data in the original publication. $\tau$ determines the time by which no nets are retained in the population, and was estimated from the assumed net half-life, as follows:

$$\tau = \frac{ITN\ half-life}{\sqrt{1 - \dfrac{\kappa}{\kappa - \ln\left(0.5\right)}}}$$

Integrating the net loss function over a distribution cycle then allows to derive the annual nets distributed per capita:

$$Nets\ distributed\ per\ capita\ per\ year = \frac{nets\ per\ capita}{DF * \int_0^{DF} p\left(t\right)\ dt}$$

Where $DF$ represents the distribution frequency.

**Appendix 1—table 4.** Parameter values for the insecticide-treated net (ITN) costing conversion.

| Parameter | Symbol | Value | Source |
|---|---|---|---|
| ITN usage | - | Varies in simulations | - |

*Appendix 1—table 4 Continued on next page*

*Appendix 1—table 4 Continued*

| Parameter | Symbol | Value | Source |
|---|---|---|---|
| ITN use rate (proportion) | - | 0.84 | Median across African countries in 2019 (**Bertozzi-Villa et al., 2021**) |
| ITN half-life (years) | - | 1.64 | Median across African countries in 2020 (**Bertozzi-Villa et al., 2021**) |
| ITN distribution frequency (years) | *DF* | 3 | World Malaria Report (**World Health Organization, 2007**; **World Health Organization, 2020**) |
| Net loss function rate parameter | $\kappa$ | 20 | **Bertozzi-Villa et al., 2021** |

## Optimization model

The mathematical problem consists of finding the allocation *b* of ITNs that minimizes global malaria cases, i.e., the sum of the product between the population $p_i$ times the clinical incidence $cinc_i$ for each EIR setting *i*. In the objective function, we also allow for the option of placing a positive contribution on settings reaching a pre-elimination phase (defined as a clinical incidence of less than 1 case per 1000 persons at risk) in addition to minimizing the global malaria case burden. This premium accounts for the potential benefits of reaching low levels of malaria transmission that go beyond the reduction in cases, e.g., general health system strengthening. For each setting reaching pre-elimination, the total remaining cases are reduced by a proportion *w* of the total cases averted by the ITN allocation (compared to total cases at baseline/without interventions), *C*. *w*, therefore, represents the weighting placed on pre-elimination in a setting relative to total case reduction. In the scenario optimized for case reduction, this weight equals 0.

This optimization must respect the budget constraint that the cost of ITNs distributed at each EIR setting $b_i$ must be less than or equal to the total budget *B*, with *c* being the cost of a single pyrethroid-treated net. In addition, the ITN usage $b_i^*$ in each setting *i* must be between 0% and an upper limit of 80%, which is a common target for universal access (**Koenker et al., 2018**). Notice that in our model, ITN distributed $b_i$ is not the same as ITN usage $b_i^\star$, because only a fraction of ITNs distributed will be used over time. We represent with $f(b_i)$ the function that maps ITNs distributed into ITNs used (see 'Relationship between distribution and usage of ITNs' for more details on this function):

$$\min_{b \in R^n} \left[ \sum_{i}^{n} cinc_i * p_i \;-\; w * C * \sum_{i=1}^{n} j_i \right]$$

$$\text{s.t.} \qquad \sum_{i=1}^{n} b_i * c \leq B$$

$$0 \leq b_i^\star \leq 0.8 \;\; \forall\, i = 1, \ldots, n$$

$$C = Cases\ at\ baseline - \sum_{i}^{n} cinc_i * p_i$$

$$j_i = \begin{cases} c1, & cinc_i < 1/1000 \\ 0, & cinc_i \geq 1/1000 \end{cases}$$

$$b_i^\star = f(b_i)$$

$$\text{for all } i = 1, \ldots, n$$

Optimization was performed using generalized simulated annealing using the *GenSA* R package (v.1.1.7.) (**Xiang et al., 2013**). *GenSA* can receive a non-linear objective function and searches an inputted search space for the global minimum. The function can tolerate a field which contains multiple local minima by simulating an annealing process using the stochasticity of a temperature parameter to escape local minima and continue the search for a global minimum (**Xiang et al., 2013**). Because many different combinations of ITN usage levels across different settings can lead to small case numbers, our objective function has many local minima. Therefore, we decided, as suggested in **Xiang et al., 2013**, to use a high value of $10^6$ for the temperature and to increase the maximum number of iterations from the default values of $5 * 10^4$–$5 * 10^6$.

Since this version of the algorithm is not designed for constrained optimization, we transformed the problem into an unconstrained optimization by introducing a penalty term in the objective function. The unconstrained problem without the pre-elimination premium can be represented as:

$$\min_{b \in R^n} \sum_{i=1}^{n} cinc_i \cdot p_i + F(b)$$

$$\text{s.t.} \qquad 0 \le b_i^\star \le 0.8 \ \ \forall \, i = 1, \ldots, n$$

$$\text{with} \quad F(b) = \begin{cases} 0 & \text{if } \sum_i b_i * c \le B \\ 10^{10} & \text{if } \sum_i b_i * c > B \end{cases}$$

$$\text{and} \qquad b_i^\star = f(b_i)$$

Namely, the objective function will assume a very high value in all cases where the budget constraint is not respected. In this way, the simulated annealing algorithm would discard all solutions outside of the budgetary constraints.

The search space was built using the *Akima* package (v.0.6–2.2, **Akima et al., 2022**), to construct two 3D surfaces of clinical incidence model outputs for every combination of bed net usage and EIR (**Appendix 1—figure 4**). The dimensions of the resulting surfaces were 9000 × 9000 points.

The optimization function was run through a range of $B$ from no intervention (starting point) to full coverage, with results indicating the resource allocation combination which most reduced clinical incidence from baseline at each level of funding.

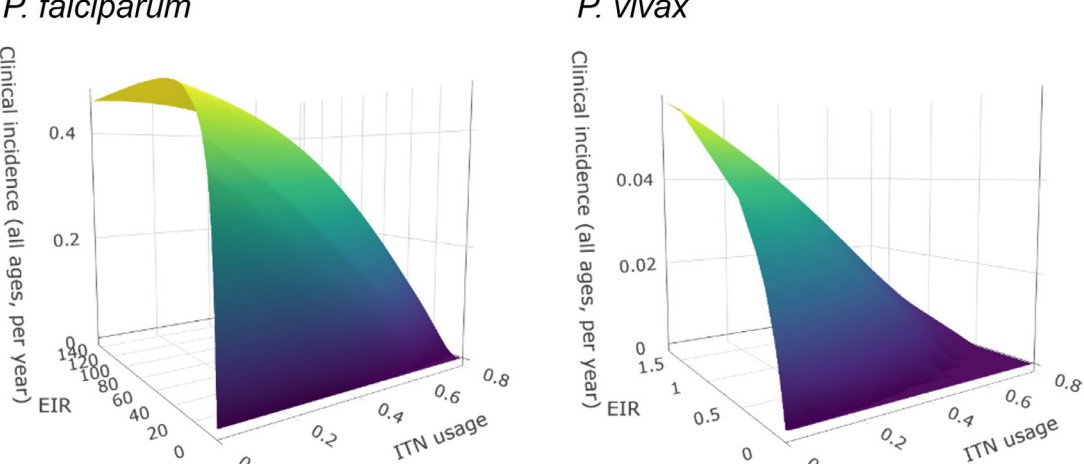

**Appendix 1—figure 4.** Surface plots of entomological inoculation rate (EIR) vs. insecticide-treated net (ITN) usage vs. clinical incidence. Plots were fit using bivariate linear interpolation of gridded data EIR and ITN usage values taken from mathematical model simulations.

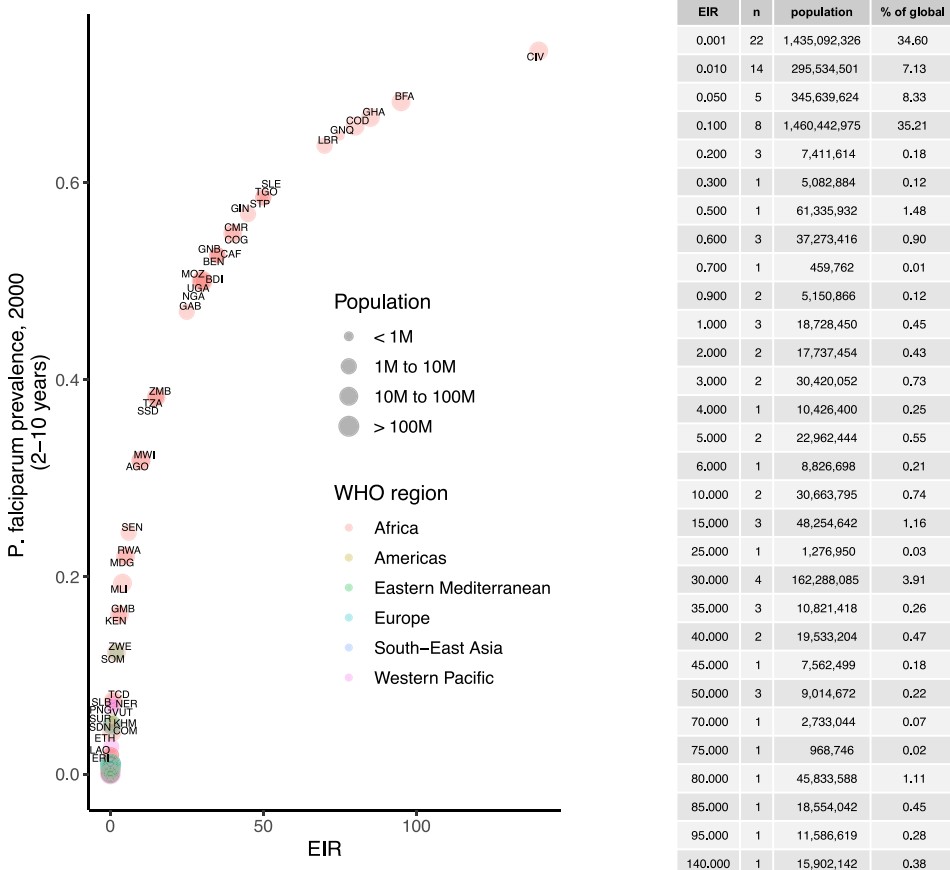

| EIR | n | population | % of global |
|---|---|---|---|
| 0.001 | 22 | 1,435,092,326 | 34.60 |
| 0.010 | 14 | 295,534,501 | 7.13 |
| 0.050 | 5 | 345,639,624 | 8.33 |
| 0.100 | 8 | 1,460,442,975 | 35.21 |
| 0.200 | 3 | 7,411,614 | 0.18 |
| 0.300 | 1 | 5,082,884 | 0.12 |
| 0.500 | 1 | 61,335,932 | 1.48 |
| 0.600 | 3 | 37,273,416 | 0.90 |
| 0.700 | 1 | 459,762 | 0.01 |
| 0.900 | 2 | 5,150,866 | 0.12 |
| 1.000 | 3 | 18,728,450 | 0.45 |
| 2.000 | 2 | 17,737,454 | 0.43 |
| 3.000 | 2 | 30,420,052 | 0.73 |
| 4.000 | 1 | 10,426,400 | 0.25 |
| 5.000 | 2 | 22,962,444 | 0.55 |
| 6.000 | 1 | 8,826,698 | 0.21 |
| 10.000 | 2 | 30,663,795 | 0.74 |
| 15.000 | 3 | 48,254,642 | 1.16 |
| 25.000 | 1 | 1,276,950 | 0.03 |
| 30.000 | 4 | 162,288,085 | 3.91 |
| 35.000 | 3 | 10,821,418 | 0.26 |
| 40.000 | 2 | 19,533,204 | 0.47 |
| 45.000 | 1 | 7,562,499 | 0.18 |
| 50.000 | 3 | 9,014,672 | 0.22 |
| 70.000 | 1 | 2,733,044 | 0.07 |
| 75.000 | 1 | 968,746 | 0.02 |
| 80.000 | 1 | 45,833,588 | 1.11 |
| 85.000 | 1 | 18,554,042 | 0.45 |
| 95.000 | 1 | 11,586,619 | 0.28 |
| 140.000 | 1 | 15,902,142 | 0.38 |

**Appendix 1—figure 5.** Prevalence of *P. falciparum* in children 2–10 years (2000), matched to entomological inoculation rate (EIR) values by country. Points are sized by total population and colored by World Health Organization region.

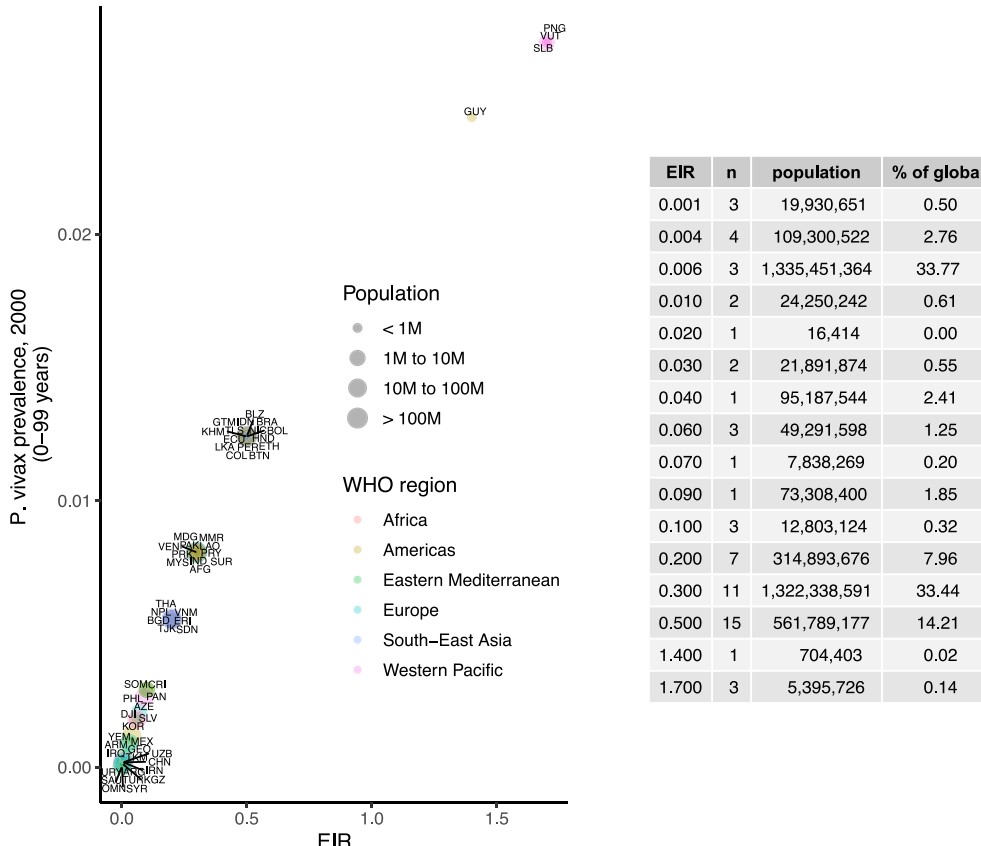

| EIR | n | population | % of global |
|---|---|---|---|
| 0.001 | 3 | 19,930,651 | 0.50 |
| 0.004 | 4 | 109,300,522 | 2.76 |
| 0.006 | 3 | 1,335,451,364 | 33.77 |
| 0.010 | 2 | 24,250,242 | 0.61 |
| 0.020 | 1 | 16,414 | 0.00 |
| 0.030 | 2 | 21,891,874 | 0.55 |
| 0.040 | 1 | 95,187,544 | 2.41 |
| 0.060 | 3 | 49,291,598 | 1.25 |
| 0.070 | 1 | 7,838,269 | 0.20 |
| 0.090 | 1 | 73,308,400 | 1.85 |
| 0.100 | 3 | 12,803,124 | 0.32 |
| 0.200 | 7 | 314,893,676 | 7.96 |
| 0.300 | 11 | 1,322,338,591 | 33.44 |
| 0.500 | 15 | 561,789,177 | 14.21 |
| 1.400 | 1 | 704,403 | 0.02 |
| 1.700 | 3 | 5,395,726 | 0.14 |

**Appendix 1—figure 6.** Prevalence of *P. vivax* in people 0–99 years (2000), matched to entomological inoculation rate (EIR) values by country. Points are sized by total population and colored by World Health Organization region.

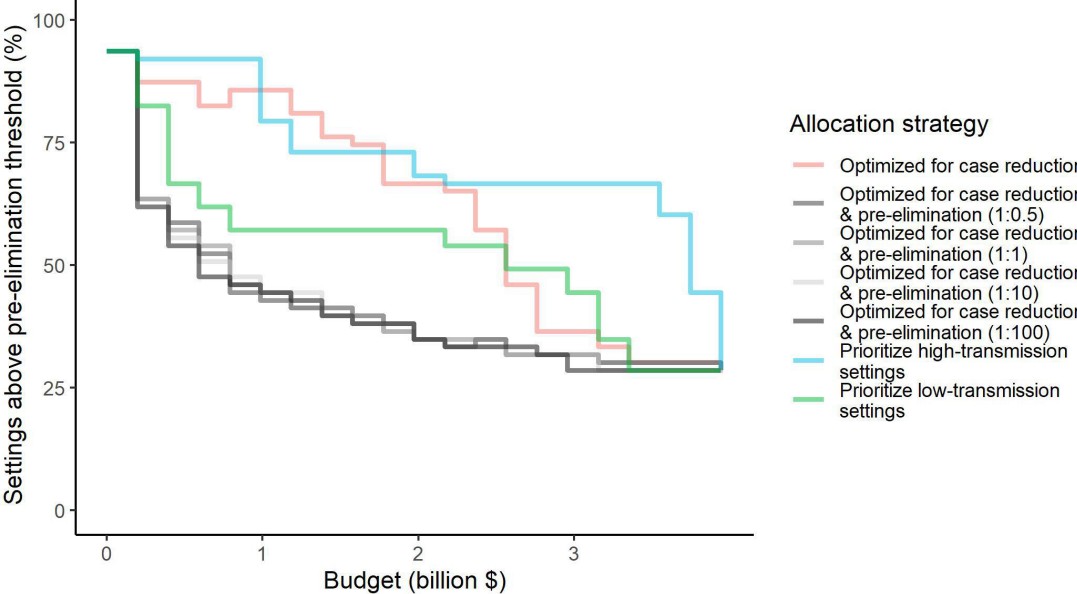

**Appendix 1—figure 7.** Percentage of settings not having reached pre-elimination (<1 case per 1000 population at risk) under different allocation strategies at varying budgets. Budget levels range from 0, representing no usage of insecticide-treated nets, to the budget required to achieve the maximum possible impact. For the strategies *Appendix 1—figure 7 continued on next page*

*Appendix 1—figure 7 continued*
optimizing for case reduction and pre-elimination, brackets show the weight placed on averting total cases *vs* on reaching pre-elimination in the optimization (i.e. 1:1 represents equal weight on both).

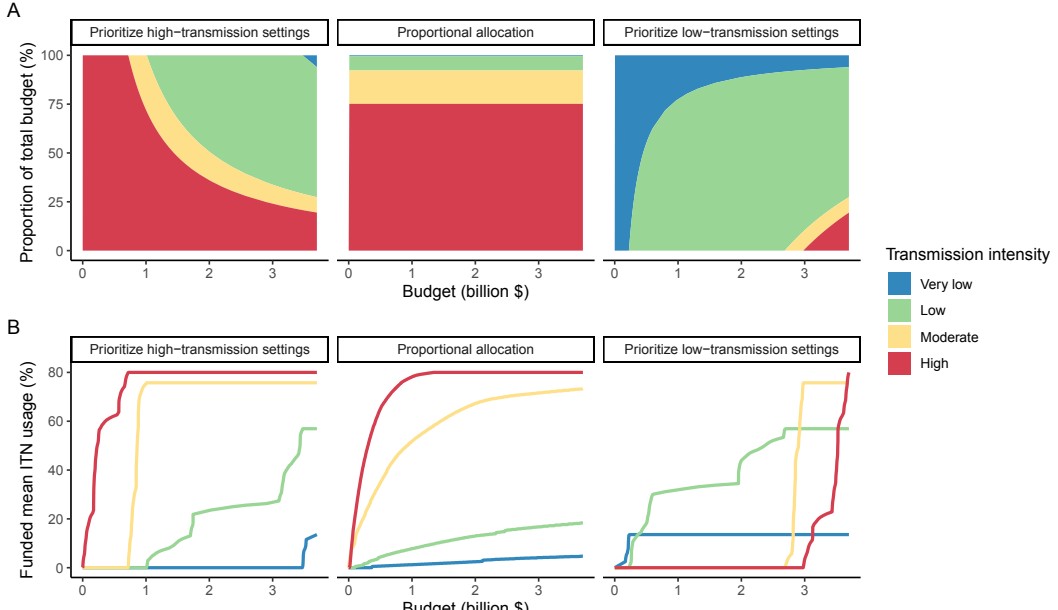

**Appendix 1—figure 8.** Illustration of resource allocation patterns under the three modeled policy strategies at varying budgets. (**A**) Percentage of global funding allocated to each transmission setting. (**B**) Mean funded insecticide-treated net (ITN) usage in each transmission setting. Transmission intensity groups represent transmission settings with varying population sizes proportional to the global distribution.

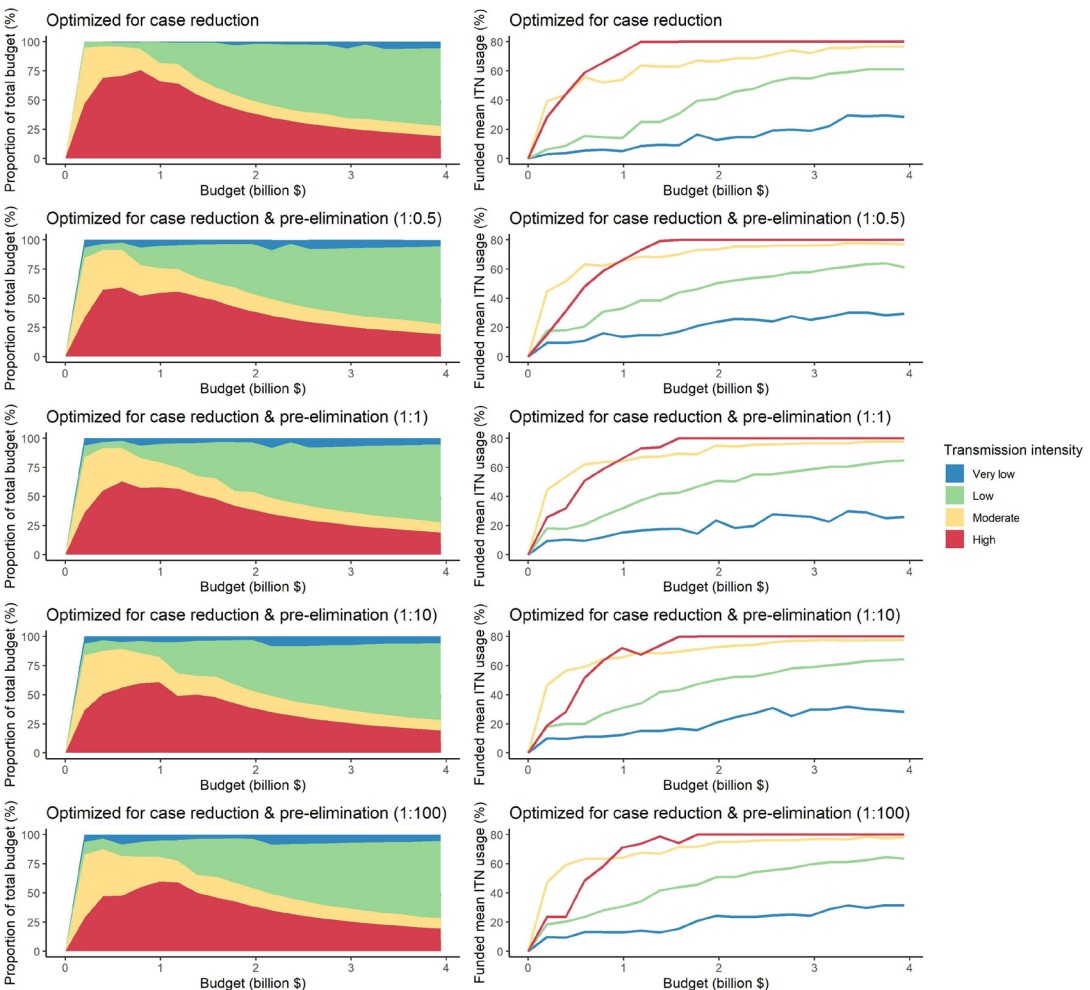

**Appendix 1—figure 9.** Optimal strategies for funding allocation across settings to minimize malaria case burden (top panels) and to minimize malaria cases and increase the number of settings reaching a pre-elimination phase at varying budgets. Panels show the proportion of the budget allocated and the resulting mean population usage of insecticide-treated nets (ITNs) across settings of different transmission intensities. For the strategies optimizing for case reduction and pre-elimination, brackets show the weight placed on averting total cases *vs* on reaching pre-elimination in the optimization (i.e. 1:1 represents equal weight on both).

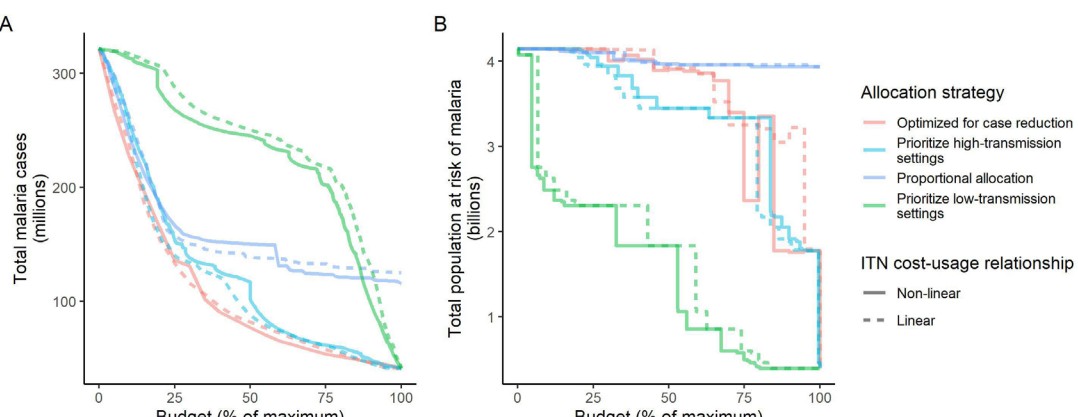

**Appendix 1—figure 10.** Influence of different assumptions about the relationship between the cost and population usage of insecticide-treated nets (ITNs) on the impact of the allocation strategies. The global clinical malaria cases (panel **A**) and the population at risk of malaria (panel **B**) under different allocation strategies are

*Appendix 1—figure 10 continued*

shown at varying budgets. Results with the more realistic non-linear assumption are presented throughout the main manuscript. Budget levels are expressed relative to the maximum budget required to achieve the largest possible impact with ITNs, but note that this maximum budget was different depending on the ITN costing assumption ($3,698,727,241 for non-linear *vs.* $6,902,923,309 for linear).

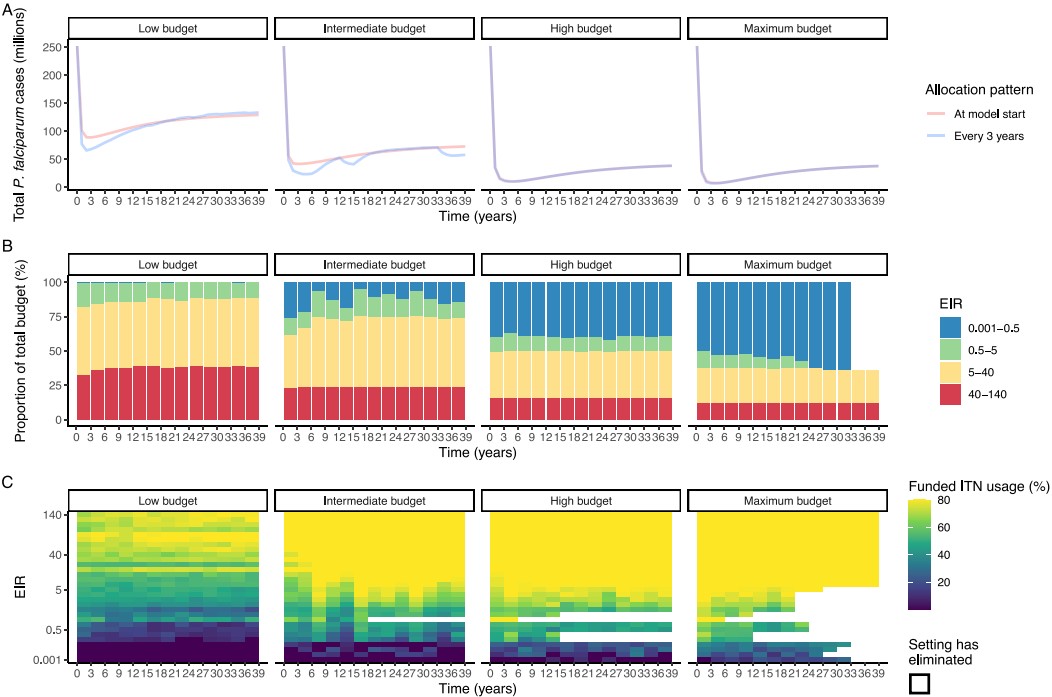

**Appendix 1—figure 11.** Resource allocation patterns over time for *P. falciparum*. Panel **A** shows the number of cases over time for re-allocation of insecticide-treated nets (ITNs) every 3 years compared to the one-time allocation of a constant ITN usage to minimize the final (year 39) case burden. Panels **B** and **C** show the optimal allocation pattern for each 3 year distribution cycle across settings of different transmission intensities. The maximum budget was 26.6 million for ITN distributions every 3 years over 39 years.

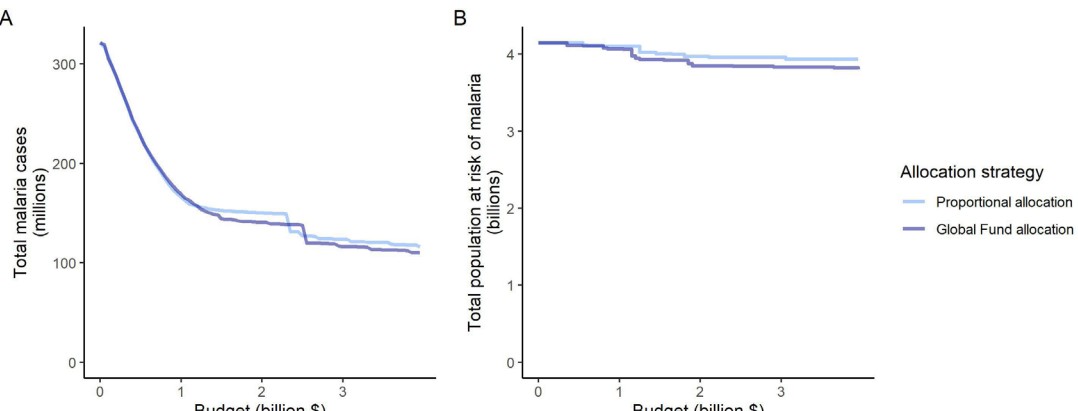

**Appendix 1—figure 12.** Impact of the proportional allocation strategy and the 2020–2022 Global Fund allocation on global malaria cases (panel **A**) and the total population at risk of malaria (panel **B**) at varying budgets. Both strategies use the same algorithm for budget share allocation based on the malaria disease burden in 2000–2004, but the Global Fund allocation additionally involves an economic capacity component and specific strategic priorities (***The Global Fund, 2023***).

