## [Editor Report · eLife assessment]

This study presents a **valuable** finding on the optimal prioritization in different malaria transmission settings for the distribution of insecticide-treated nets to reduce the malaria burden. The evidence supporting the claims of the authors is **solid**. The work will be of interest from a global funder perspective, though somewhat less relevant for individual countries.

---

## [Referee Report · Reviewer #1 (Public Review)]

Schmit et al. analyze and compare different strategies for the allocation of funding for insecticide-treated nets (ITNs) to reduce the global burden of malaria. They use previously published models of *Plasmodium falciparum* and *Plasmodium vivax* malaria transmission to quantify the effect of ITN distribution on clinical malaria numbers and the population at risk. The impact of different resource allocation strategies on the reduction of malaria cases or a combination of malaria cases and achieving pre-elimination is considered to determine the optimal strategy to allocate global resources to achieve malaria eradication.

Strengths:

Schmit et al. use previously published models and optimization for a rigorous analysis and comparison of the global impact of different funding allocation strategies for ITN distribution. This provides evidence of the effect of three different approaches: the prioritization of high-transmission settings to reduce the disease burden, the prioritization of low-transmission settings to "shrink the malaria map", and a resource allocation proportional to the disease burden.

Weaknesses:

The analysis and optimization which provide the evidence for the conclusions and are thus the central part of this manuscript necessitate some simplifying assumptions which may have important practical implications for the allocation of resources to reduce the malaria burden. For example, seasonality, mosquito species-specific properties, stochasticity in low transmission settings, and changing population sizes were not included. Other challenges to the reduction or elimination of malaria such as resistance of parasites and mosquitoes or the spread of different mosquito species as well as other beneficial interventions such as indoor residual spraying, seasonal malaria chemoprevention, vaccinations, combinations of different interventions, or setting-specific interventions were also not included. Schmit et al. clearly state these limitations throughout their manuscript.

This work considers different ITN distribution strategies, other interventions are not considered. It also provides a global perspective but an analysis of the specific local setting (as also noted by Schmit et al.) and different interventions as well as combinations of interventions should also be taken into account for any decisions. Nonetheless, the rigorous analysis supports the authors' conclusions and provides evidence that supports the prioritization of funding of ITNs for settings with high *Plasmodium falciparum* transmission. Overall, this work may contribute to making evidence-based decisions regarding the optimal prioritization of funding and resources to achieve a reduction in the malaria burden.

---

## [Referee Report · Reviewer #2 (Public Review)]

Summary:

In this article, the authors discuss an optimal resource allocation strategy to best allocate funding in maximising malaria eradication efforts. Though achieving elimination by only using insecticide-treated bed nets (ITNs) is not the best practice, and countries utilise different interventions simultaneously, this analysis could be relevant in allocating funding for the global malaria elimination effort. To analyse and compare the impact of ITNs on *P. falciparum* and *P. vivax* cases and the total populations at risk, the authors use two previously published models (for *P. falciparum* and *P. vivax*).

Strengths:

The authors use models for both *P. falciparum* and *P. vivax* to analyse the impact of different strategies for allocating ITNs and provide the best strategies for funding to minimise malaria burden across different transmission settings. Using previously published models that account for various malaria aspects, including demography, heterogeneity in bite exposure, immunity, variation in hypnozoite across bites (*P. vivax*), mosquito larval dynamics, etc., gives a solid foundation for the analysis performed here.

Weaknesses:

Though the objective of the study is to identify the best setting to allocate funding to eradicate malaria, the authors use prevalence estimates (*P. falciparum* and *P. vivax*) based on the year 2000 as the baseline. Given their reasoning behind this choice, the analysis would be more relevant or useful if the proposed strategy were compared to the current Global Technical Strategy for Malaria (GTS 2016-2030). That is, using estimates based on around the year 2016.

In settings where both *P. falciparum* and *P. vivax* are co-endemic, using models that do not account for the interplay between the species, especially regarding immunity, somewhat underplays the overall disease dynamics. Furthermore, assuming the transmission within each setting (very low, low, moderate, high) is homogenous is also a weakness as there is heterogeneity in transmission intensity, bite exposure, etc, within each setting.

---

## [Author Response]

The following is the authors’ response to the current reviews.

We thank the editors and reviewers for their helpful comments, which have allowed us to improve the manuscript.

**Response to reviewer 2**

We thank the reviewer for this positive feedback, which requires no further revision.

**Response to reviewer 3**

We thank the reviewer for highlighting these additional points and provide further explanations on these below.

Firstly, we started the analysis from a baseline of year 2000 because the largest international donor (the Global Fund) uses baseline malaria levels in the period 2000-2004 as the basis of their current allocation calculations (The Global Fund, Description of the 2020-2022 Allocation Methodology, December 2019). In the paper we compare our optimal strategy to a simplified version of this method, represented by our “proportional allocation” strategy.

Even if our simulations started in the year 2015, a direct comparison with the Global Technical Strategy for Malaria 2016-2030 would not be possible due to the different approaches taken. The GTS was developed to progress towards malaria elimination globally and set ambitious targets of at least 90% reduction in malaria case incidence and mortality rates and malaria elimination in at least 35 countries by 2030 compared to 2015. Mathematical modelling at the time suggested that 90% coverage of WHO-recommended interventions (vector control, treatment and seasonal malaria chemoprevention) would be needed to approach this target (Griffin et al. 2016, Lancet Infectious Diseases). The global annual investment requirements to meet GTS targets were estimated at US$6.4 billion by 2020 and US$8.7 billion by 2030 (Patouillard et al. 2017, BMJ Global Health). This strategy therefore considers what resources would be required to achieve a specific global target, but not the optimized allocation of resources.

Investments into malaria control have consistently been below the estimated requirements for the GTS milestones (World Health Organization 2022, World Malaria Report 2022). In our study, we therefore take a different perspective on how limited budgets can be optimally allocated to a single intervention (insecticide-treated nets) across countries/settings to achieve the best possible outcome for two objectives that are different to the GTS milestones (either minimizing the global case burden, or minimizing both the global case burden and the number of settings not having yet reached a pre-elimination phase). As stated in the discussion, our estimate of allocating 76% of very low budgets to high-transmission settings was similar to the global investment targets estimated for the GTS, where the 20 countries with the highest burden in 2015 were estimated to require 88% of total investments (Patouillard et al. 2017, BMJ Global Health). Nevertheless, we also show that if higher budgets were available, allocating the majority to low-transmission settings co-endemic for *P. falciparum* and *P. vivax* would achieve the largest reduction in global case burden. We acknowledge the modelling of a single intervention as one of the key limitations of this analysis, but this simplification was necessary in order to perform the complex optimisation problem. Computationally it would not have been feasible to optimize across a multitude of intervention and coverage combinations.

A further limitation raised by the reviewer is the lack of cross-species immunity between *P. falciparum* and *P. vivax* in our model. While cross-reactivity between antibodies against these two species has been observed in previous studies and the potential implications of this would be important to explore in future work, we did not include it here as little is known to date about the epidemiological interactions between different malaria parasite species (Muh et al. 2020, PLoS Neglected Tropical Diseases).

Lastly, we did not assume that transmission was homogenous within the four transmission settings in our study (very low, low, moderate, high); transmission dynamics were simulated separately in each country, accounting for heterogeneous mosquito bite exposure. However, results were summarised for the broader transmission settings since many other country-specific factors were not accounted for (see discussion) and the findings should not be used to inform individual country allocation decisions.

The following is the authors’ response to the original reviews.

Author response to peer review

We thank the reviewers for their insightful comments, which raise several important points regarding our study. As the reviewers have recognised, we introduced a number of simplifications in order to perform this complex optimisation problem, such as by restricting the analysis to a single intervention (insecticide-treated nets) and modelling countries at a national level. Despite their clear relevance to the study, computationally it would not have been feasible to run the multitude of scenarios suggested by reviewer 1, which we recognise as a limitation. As such we agree with the assessment that this study primarily represents a thought experiment, based on substantive modelling and aggregate scenario-based analysis, to assess whether current policies are aligned with an optimal allocation strategy or whether there might be a need to consider alternative strategies. The findings are relevant primarily to global funders and should not be used to inform individual country allocation decisions, and also point to avenues for further research. This perspective also underlies our decision to start the analysis from a baseline of year 2000 as opposed to modelling the current 2023 malaria situation: the largest international donor (the Global Fund) uses baseline malaria levels in the period 2000-2004 as the basis of their allocation calculations (The Global Fund, Description of the 2020-2022 Allocation Methodology, December 2019) (1). A simplified version of this method is represented by our “proportional allocation” strategy. We have made several revisions to the manuscript to address the points raised by the reviewers, as detailed below.

**Reviewer #1 (Public Review):**
1. The authors present a back-of-the-envelope exploration of various possible resource allocation strategies for ITNs. They identify two optimal strategies based on two slightly different objective functions and compare 3 simple strategies to the outcomes of the optimal strategies and to each other. The authors consider both P falciparum and P vivax and explore this question at the country level, using 2000 prevalence estimates to stratify countries into 4 burden categories. This is a relevant question from a global funder perspective, though somewhat less relevant for individual countries since countries are not making decisions at the global scale.

Thank you for this summary of the paper. We agree that our analysis is of relevance to global funders, but is not meant to inform individual country allocation decisions. In the discussion, we now state:

p. 12 L19: “Therefore, policy decisions should additionally be based on analysis of country-specific contexts, and our findings are not informative for individual country allocation decisions.”

1. The authors have made various simplifications to enable the identification of optimal strategies, so much so that I question what exactly was learned. It is not surprising that strategies that prioritize high-burden settings would avert more cases.

Thank you for raising this point. Indeed, several simplifying assumptions were necessary to ensure the computational feasibility of this complex optimization problem. As a result, our study primarily represents a thought experiment to assess whether current policies are aligned with an optimal allocation strategy or whether there might be a need to consider alternative strategies. As now further outlined in the introduction, approaches to this have differed over time and it remains a relevant debate for malaria policy.

p. 2 L22: “However, there remains a lack of consensus on how best to achieve this longer-term aspiration. Historically, large progress was made in eliminating malaria mainly in lower-transmission countries in temperate regions during the Global Malaria Eradication Program in the 1950s, with the global population at risk of malaria reducing from around 70% of the world population in 1950 to 50% in 2000 (2). Renewed commitment to malaria control in the early 2000s with the Roll Back Malaria initiative subsequently extended the focus to the highly endemic areas in sub-Saharan Africa (3).”

We believe our findings not only confirm an “expected” outcome – that prioritizing high-burden settings would avert more cases – but also clearly illustrate various consequences of different allocation strategies that are implemented or considered in reality, which may not be so obvious. For example, we found that initially allocating a larger share of the budget to high-transmission countries could be both almost optimal in terms of reducing clinical cases and maximising the number of countries reaching pre-elimination. We also observed a trade-off between reducing burden and reducing the global population at risk (“shrinking the map”) through a focus on near-elimination settings, and estimate the loss in burden reduction when following an elimination target.

1. Generally, I found much of the text confusing and some concepts were barely explained, such that the logic was difficult to follow.

Thank you for bringing this to our attention, and we regret to hear the manuscript was confusing to read. We believe that the revisions made as a result of the reviewer comments have now made the manuscript much easier to follow. We additionally passed the manuscript to a colleague to identify confusing passages, and have added a number of sentences to clarify key concepts and improve the structure.

1. I am not sure why the authors chose to stratify countries by 2000 PfPR estimates and in essence explore a counterfactual set of resource allocation strategies rather than begin with the present and compare strategies moving forward. I would think that beginning in 2020 and modeling forward would be far more relevant, as we can't change the past. Furthermore, there was no comparison with allocations and funding decisions that were actually made between 2000 and 2020ish so the decision to begin at 2000 is rather confusing.

Thank you for pointing this out. We have now made the rationale for this choice clearer in the manuscript. Our main reason for this was to allow comparison with the Global Fund funding allocation, which is largely based on malaria disease burden in 2000-2004. As stated in the paper, malaria prevalence estimates in the year 2000 are commonly considered to represent a “baseline” endemicity level, before large-scale implementation of interventions in the following decades. In the manuscript, the transmission-related element of the Global Fund allocation algorithm is represented in our “proportional allocation” strategy. Previously this was only mentioned in the methods, but we have now added the following in the results to address this comment of the reviewer:

p. 6 L12: “Strategies prioritizing high- or low-transmission settings involved sequential allocation of funding to groups of countries based on their transmission intensity (from highest to lowest EIR or vice versa). The proportional allocation strategy mimics the current allocation algorithm employed by the Global Fund: budget shares are mainly distributed according to malaria disease burden in the 2000-2004 period. To allow comparison with this existing funding model, we also started allocation decisions from the year 2000.”

The Global Fund framework additionally considers economic capacity and other specific factors, and we have now also included a direct comparison with the 2020-2022 Global Fund allocation in Supplementary Figure S12 (see Author response image 1).

We agree that looking at allocation decisions from 2020 onward would also constitute a very interesting question. However, the high dimensionality in scenarios to consider for this would currently make it computationally infeasible to run on the global level. Not only would it have to include all interventions currently implemented and available for malaria at different levels of coverage, but also the option of scaling down existing interventions. Instead, our priority in this paper was to conduct a thought experiment including both *P. falciparum* and *P. vivax* on a large geographical scale.

**Author response image 1. sa3fig1:** Impact of the proportional allocation strategy and the 2020-2022 Global Fund allocation on global malaria cases (panel A) and the total population at risk of malaria (panel B) at varying budgets. Both strategies use the same algorithm for budget share allocation based on malaria disease burden in 2000-2004, but the Global Fund allocation additionally involves an economic capacity component and specific strategic priorities.

1. I realize this is a back-of-the-envelope assessment (although it is presented to be less approximate than it is, and the title does not reveal that the only intervention strategy considered is ITNs) but the number and scope of modeling assumptions made are simply enormous. First, that modeling is done at the national scale, when transmission within countries is incredibly heterogeneous. The authors note a differential impact of ITNs at various transmission levels and I wonder how the assumption of an intermediate average PfPR vs modeling higher and lower PfPR areas separately might impact the effect of the ITNs.

Thank you for this comment. We agree the title could be more specific and have changed this to “Resource allocation strategies for insecticide-treated bednets to achieve malaria eradication”.

Regarding the scale of ITN allocation, it is true that allocation at a sub-national scale could affect the results. However, considering this at a national scale is most relevant for our analysis because this is the scale at which global funding allocation decisions are made in practice. A sentence explaining this has been added in the methods.

p. 15 L8: “The analysis was conducted on the national level, since this scale also applies to funding decisions made by international donors (1).”

Further considering different geographical scales would also require introducing other assumptions, for example about how different countries would distribute funding sub-nationally, whether specific countries would take cooperative or competitive approaches to tackle malaria within a region or in border areas, and about delays in the allocation of bednets in specific regions. These interesting questions were outside of the scope of this work, but certainly require further investigation.

1. Second, the effect of ITNs will differ across countries due to variations in vector and human behavior and variation in insecticide resistance and susceptibility to the ITNs. The authors note this as a limitation but it is a little mind-boggling that they chose not to account for either factor since estimates are available for the historical period over which they are modeling.

Thank you for pointing this out. We did consider this and mentioned it as a limitation. Nevertheless, the complexity of accounting for this should also be recognised; for example, there is substantial uncertainty about the precise relationship between insecticide resistance and the population-level effect of ITNs (Sherrard-Smith et al., 2022, Lancet Planetary Health) (4). Additionally, our simulations extend beyond the 2000-2023 period so further assumptions about future changes to these factors would also be required. Simplifying assumptions are inherent to all mathematical modelling studies and we consider these particular simplifications acceptable given the high-level nature of the analysis.

1. Third, the assumption that elimination is permanent and nothing is needed to prevent resurgence is, as the authors know, a vast oversimplification. Since resources will be needed to prevent resurgence, it appears this assumption may have a substantial impact on the authors' results.

Thank you for this comment. In the discussion, we have now expanded on this:

p. 13 L3: “While our analysis presents allocation strategies to progress towards eradication, the results do not provide insight into allocation of funding to maintain elimination. In practice, the threat of malaria resurgence has important implications for when to scale back interventions.”

We believe that from a global perspective, the questions of funding allocation to achieve elimination vs to maintain it can currently still be considered separately given the large time-scales involved. The cost of preventing resurgence is not known, and one major problem in accounting for this would also be to identify relevant timescales to quantify this over.

1. The decision to group all settings with EIR > 7 together as "high transmission" may perhaps be driven by WHO definitions but at a practical level this groups together countries with EIR 10 and EIR 500. Why not further subdivide this group, which makes sense from a technical perspective when thinking about optimal allocation strategies?

Thank you for pointing this out. The WHO categories used are better interpreted in terms of the corresponding prevalence, which places countries with a prevalence of over 35% in the high transmission categories (WHO Guidelines for malaria, 31 March 2022) (5). We felt this is appropriate given that we are looking at theoretical global allocation patterns and do not aim to make recommendations for specific groups of countries or individual countries within sub-Saharan Africa that would be distinguished through the use of higher cut-offs. In our analysis, all 25 countries in the high transmission category were located in sub-Saharan Africa.

1. The relevance of this analysis for elimination is a little questionable since no one eliminates with ITNs alone, to the best of my understanding.

Thank you for this comment. We indeed state in the paper that ITNs alone are not sufficient to eliminate malaria. However, we still think that our analysis is relevant for elimination by taking a more theoretical perspective on reducing transmission using interventions. Starting from the 2000 baseline (or current levels) globally, large-scale transmission reductions such as those achieved by mass ITN distribution still represent the first key step on the path to malaria eradication, as shown in previous modelling work (Griffin et al., 2016, Lancet Infectious Diseases) (6). In the final phase of elimination, the WHO also recommends the addition of more targeted and reactive interventions (WHO Guidelines for malaria, 31 March 2022) (5). Our changes to the title of the article (“Resource allocation strategies for insecticide-treated bednets to achieve malaria eradication”) should now better reflect that we consider ITNs as just one necessary component to achieve malaria eradication.

**Reviewer #2 (Public Review):**
1. Schmit et al. analyze and compare different strategies for the allocation of funding for insecticide-treated nets (ITNs) to reduce the global burden of malaria. They use previously published models of *Plasmodium falciparum* and Plasmodium vivax malaria transmission to quantify the effect of ITN distribution on clinical malaria numbers and the population at risk. The impact of different resource allocation strategies on the reduction of malaria cases or a combination of malaria cases and achieving pre-elimination is considered to determine the optimal strategy to allocate global resources to achieve malaria eradication.Strengths:Schmit et al. use previously published models and optimization for rigorous analysis and comparison of the global impact of different funding allocation strategies for ITN distribution. This provides evidence of the effect of three different approaches: the prioritization of high-transmission settings to reduce the disease burden, the prioritization of low-transmission settings to "shrink the malaria map", and a resource allocation proportional to the disease burden.

Thank you for providing this summary and outline of the strengths of the paper.

1. Weaknesses:The analysis and optimization which provide the evidence for the conclusions and are thus the central part of this manuscript necessitate some simplifying assumptions which may have important practical implications for the allocation of resources to reduce the malaria burden. For example, seasonality, mosquito species-specific properties, stochasticity in low transmission settings, and changing population sizes were not included. Other challenges to the reduction or elimination of malaria such as resistance of parasites and mosquitoes or the spread of different mosquito species as well as other beneficial interventions such as indoor residual spraying, seasonal malaria chemoprevention, vaccinations, combinations of different interventions, or setting-specific interventions were also not included. Schmit et al. clearly state these limitations throughout their manuscript.The focus of this work is on ITN distribution strategies, other interventions are not considered. It also provides a global perspective and analysis of the specific local setting (as also noted by Schmit et al.) and different interventions as well as combinations of interventions should also be taken into account for any decisions.

Thank you for raising these points. As outlined at the beginning of our response, for computational reasons we indeed had to introduce several simplifying assumptions to perform this complex optimisation problem. As a result of these factors you highlighted, our study should primarily be interpreted as a thought experiment to assess whether current policies are aligned with an optimal allocation strategy or whether there might be a need to consider alternative strategies. The findings are relevant primarily to global funders and should not be used to inform individual country allocation decisions, which we have further clarified in the manuscript.

1. Nonetheless, the rigorous analysis supports the authors' conclusions and provides evidence that supports the prioritization of funding of ITNs for settings with high *Plasmodium falciparum* transmission. Overall, this work may contribute to making evidence-based decisions regarding the optimal prioritization of funding and resources to achieve a reduction in the malaria burden.

Thank you for this positive assessment of our work.

**Reviewer #1 (Recommendations For The Authors):**
1. L144: last paragraph, the focus on endemic equilibrium: I did not really understand this, when 39 years is mentioned later is that a different analysis? How are cases averted calculated in a time-agnostic endemic equilibrium analysis? Perhaps a little more detail here would be helpful.

A further explanation of this has been added in the results and methods.

p. 8 L 22: “To evaluate the robustness of the results, we conducted a sensitivity analysis on our assumption on ITN distribution efficiency. Results remained similar when assuming a linear relationship between ITN usage and distribution costs (Figure S10). While the main analysis involves a single allocation decision to minimise long-term case burden (leading to a constant ITN usage over time in each setting irrespective of subsequent changes in burden), we additionally explored an optimal strategy with dynamic re-allocation of funding every 3 years to minimise cases in the short term.”

p. 17 L25: “To ensure computational feasibility, 39 years was used as it was the shortest time frame over which the effect of re-distribution of funding from countries having achieved elimination could be observed.”

p. 18 L 9: “Global malaria case burden and the population at risk were compared between baseline levels in 2000 and after reaching an endemic equilibrium under each scenario for a given budget.”

1. L148: what is proportional allocation by disease burden and how is that different from prioritizing high-transmission settings?

Further details have been added in the text.

p. 6 L12: “Strategies prioritizing high- or low-transmission settings involved sequential allocation of funding to groups of countries based on their transmission intensity (from highest to lowest EIR or vice versa). The proportional allocation strategy mimics the current allocation algorithm employed by the Global Fund: budget shares are mainly distributed according to malaria disease burden in the 2000-2004 period. To allow comparison with this existing funding model, we also started allocation decisions from the year 2000.”

1. L198-9: did low transmission settings get the majority of funding at intermediate and maximum budgets because they have the most population (I think so, based on Fig 1)?

Yes, this is correct. We state in the results: “the optimized distribution of funding to minimize clinical burden depended on the available global budget and was driven by the setting-specific transmission intensity and the population at risk”.

1. L206: what is ITN distribution efficiency? This is not explained. What is the 39-year period? Why this duration?

Further explanations have been added in the results section, which were previously only detailed in the methods:

p. 8 L 22: “To evaluate the robustness of the results, we conducted a sensitivity analysis on our assumption on ITN distribution efficiency. Results remained similar when assuming a linear relationship between ITN usage and distribution costs (Figure S10)."

p. 17 L25: “To ensure computational feasibility, 39 years was used as it was the shortest time frame over which the effect of re-distribution of funding from countries having achieved elimination could be observed.”

1. L218: what is "no intervention with a high budget"? is this a phrasing confusion?

Yes, this has been changed.

p. 9 L14: “We estimated that optimizing ITN allocation to minimize global clinical incidence could, at a high budget, avert 83% of clinical cases compared to no intervention.”

1. L235-7: on comparing these results to previous work on the 20 highest-burden countries: is the definition of "high" similar enough across these studies that this is a relevant comparison?

We believe this is reasonably comparable, as looking at the 20 highest-burden countries encompasses almost the entire high-transmission group in our work (25 countries in total), on which the comparison is made.

1. L267-70: I didn't understand this sentence at all.

Thanks for flagging this. The sentence referred to is: “Allocation proportional to disease burden did not achieve as great an impact as other strategies because the funding share assigned to settings was constant irrespective of the invested budget and its impact, and we did not reassign excess funding in high-transmission settings to other malaria interventions.”

The previously mentioned added details on the proportional allocation strategy in the manuscript should now make this clearer, together with this clarification:

p. 11 L17: “In modelling this strategy, we did not reassign excess funding in high-transmission settings to other malaria interventions, as would likely occur in practice.”

For proportional allocation, a fixed proportion of the budget is calculated for each country based on disease burden, as described in the Global Fund allocation documentation (see Methods). However, since ITNs are the only intervention considered, this leads to a higher budget being allocated than is needed in some countries (i.e. where more funding doesn’t translate into further health gains).

1. L339 EIR range: 80 is high at the country level but areas within countries probably went as high as 500 back in 2000. How does this affect the modeled estimates of ITN impact?

The question of sub-national differences in transmission has been addressed in the public review comments. Briefly, we consider the national scale to be most relevant for our analysis because this is the scale at which global funding allocation decisions are made in practice. Although, as you correctly point out, the EIR affects ITN impact, it is not possible to conclude what the average effect of this would be on the country level without considering the following factors and introducing further assumptions on these: how would different countries distribute funding sub-nationally? Which countries would take cooperative or competitive approaches to tackle malaria within a region or in border areas? Would there be delays in the allocation of bednets in specific regions? These interesting questions were outside of the scope of this work, but certainly require further investigation.

1. L347 population size constant: births and deaths are still present, is that right? Unclear from this sentence

Yes, this is correct. Full details on the model can be found in the Supplementary Materials.

1. L370 estimating ITN distribution required to achieve simulated population usage: is this a single relationship for all of Africa? Is it based on ITNs distributed 2:1 -> % access -> % usage? So it accounts for allocation inefficiency?

Yes, this is represented by a single relationship for all of Africa to account for allocation inefficiency and is based on observed patterns across the continent and methodology developed in a previous publication (Bertozzi-Villa et al., 2021, Nature Communications) (7). Full details can be found in the Supplementary Materials (“Relationship between distribution and usage of insecticide-treated nets (ITNs)”, p. 21).

1. L375: the ITN unit cost is assumed constant across countries and time (I think, it doesn't say explicitly), is this a good assumption?

Yes, this is correct. We consider this a reasonable assumption within the scope of the paper. While delivery costs likely vary across countries, international funders usually have pooled procurement mechanisms for ITNs (The Global Fund, 2023, Pooled Procurement Mechanism Reference Pricing: Insecticide-Treated Nets).

1. L399: "single allocation of a constant ITN usage" it is not explained what exactly this means

Further explanations have been added in the manuscript.

p. 8 L24: “While the main analysis involves a single allocation decision to minimise long-term case burden (leading to a constant ITN usage over time in each setting irrespective of subsequent changes in burden), we additionally explored an optimal strategy with dynamic re-allocation of funding every 3 years to minimise cases in the short term.”

**Reviewer #2 (Recommendations For The Authors):**
1. Additionally to the public comments, the only major comment is that in this reviewer's opinion, the focus on ITNs as the only intervention should be made clearer at different places in the manuscript (e.g. in the discussion lines 303-304). Otherwise, there are only some minor comments (see below).

We have now modified the following sentence and also included this suggestion in the title (“Resource allocation strategies for insecticide-treated bednets to achieve malaria eradication”).

p. 13 L8: “Our analysis demonstrates the most impactful allocation of a global funding portfolio for ITNs to reduceglobal malaria cases.”

1. Minor comments:It may be of interest to compare the maximum budget obtained from the optimization with other estimates of required funding and actual available funding.

Thank you for this interesting suggestion. Our maximum budget estimates are similar to the required investments projected for the WHO Global Technical Strategy: US$3.7 billion for ITNs in our analysis compared to between US$6.8 and US$10.3 billion total annual resources between 2020 and 2030, of which an estimated 55% would be required for (all) vector control (US$3.7 - US$5.7 billion) (Patouillard et al., 2016, BMJ Global Health) (8). However, it is well known that current spending is far below these requirements: total investments in malaria were estimated to be about US$3.1 billion per year in the last 5 years (World Health Organization, 2022, World Malaria Report 2022) (9).

1. Line 177: should "Figure S7" be bold?

Yes, this has been corrected.

1. Line 218: what does "no intervention with high budget" mean? Should this simply be "no intervention"?

This has been changed.

p. 9 L14: “We estimated that optimizing ITN allocation to minimize global clinical incidence could, at a high budget, avert 83% of clinical cases compared to no intervention.”

1. In this reviewer's opinion it would be easier for the reader if the weighting term in the objective function would be added in the Materials and Methods section. The weighting could be added without extending the section substantially and the explanation in lines 390-393 may be easier to understand.

Thank you for this suggestion. We agree and have added this in the main manuscript.

References

1. The Global Fund. Description of the 2020-2022 Allocation Methodology 2019 [Available from: https://www.theglobalfund.org/media/9224/fundingmodel_2020-2022allocations_methodology_en.pdf].

2. Hay SI, Guerra CA, Tatem AJ, Noor AM, Snow RW. The global distribution and population at risk of malaria: past, present, and future. Lancet Infect Dis. 2004;4(6):327-36.

3. Feachem RGA, Phillips AA, Hwang J, Cotter C, Wielgosz B, Greenwood BM, et al. Shrinking the malaria map: progress and prospects. The Lancet. 2010;376(9752):1566-78.

4. Sherrard-Smith E, Winskill P, Hamlet A, Ngufor C, N'Guessan R, Guelbeogo MW, et al. Optimising the deployment of vector control tools against malaria: a data-informed modelling study. The Lancet Planetary Health. 2022;6(2):e100-e9.

5. World Health Organization. WHO Guidelines for malaria, 31 March 2022. Geneva: World Health Organization; 2022. Contract No.: Geneva WHO/UCN/GMP/ 2022.01 Rev.1.

6. Griffin JT, Bhatt S, Sinka ME, Gething PW, Lynch M, Patouillard E, et al. Potential for reduction of burden and local elimination of malaria by reducing *Plasmodium falciparum* malaria transmission: a mathematical modelling study. The Lancet Infectious Diseases. 2016;16(4):465-72.

7. Bertozzi-Villa A, Bever CA, Koenker H, Weiss DJ, Vargas-Ruiz C, Nandi AK, et al. Maps and metrics of insecticide-treated net access, use, and nets-per-capita in Africa from 2000-2020. Nature Communications. 2021;12(1):3589.

8. Patouillard E, Griffin J, Bhatt S, Ghani A, Cibulskis R. Global investment targets for malaria control and elimination between 2016 and 2030. BMJ global health. 2017;2(2):e000176.

9. World Health Organization. World malaria report 2022. Geneva: World Health Organization; 2022. Report No.: 9240064893.